# Route-and-Reason: Scaling Large Language Model Reasoning with Reinforced Model Router

## Abstract

Chain-of-thought has been proven essential for enhancing the complex reasoning abilities of Large Language Models (LLMs), but it also leads to high computational costs. Recent advances have explored the method to route queries among multiple models and proved it as a promising approach. However, previous works directly operate at the task level, i.e., assigning user queries to suitable LLMs, which does not allow hybrid LLMs to truly collaborate on finer-grained sub-tasks. Collaboration at the level of intermediate reasoning steps (thoughts) could enable more efficient coordination, but it also poses significant challenges for router scheduling, placing immense demands on the quality of task decomposition and the precision of the router. To address this, we propose **R2-Reasoner**, a novel framework centered around **a Reinforced Model Router** designed to efficiently scale LLM reasoning. This router orchestrates collaboration across 9 heterogeneous models, of whom the parameter scale ranges from less than 1B to hundreds of billions, by first breaking down a complex query into subtasks with a decomposer, and then assigning each subtask to the optimal model with a subtask allocator, balancing performance with cost. To train this router involves a two-stage alternating process for the decomposer and the allocator, integrating supervised fine-tuning with reinforcement learning to enable effective self-supervised refinement. Extensive experiments across six challenging reasoning benchmarks demonstrate that R2-Reasoner reduces API costs by 84.46% compared with state-of-the-art baselines while maintaining competitive reasoning accuracy. Our framework paves the way for the development of more scalable and efficient reasoning systems. Our code is open-source at
`https://anonymous.4open.science/r/R2_Reasoner`.

## 1 Introduction

Chain-of-Thought (CoT, (Wei et al., 2022)) reasoning has endowed large language models (LLMs) with significantly enhanced reasoning capabilities. The reasoning ability of LLMs has evolved from prompting-based sequential thoughts to reinforcement learning–driven long-chain reasoning (OpenAI, 2024; Guo et al., 2025; Lightman et al., 2023; Snell et al., 2024; Wang et al., 2023; Chen et al., 2024b), developing into *test-time scaling* as a paradigm, though with significant computational cost (Wu et al., 2024). To mitigate the vast increase in computational overhead, **model router** has been introduced to route queries across models according to problem difficulty, model capability and associated cost. This strategy is recognized as an effective means of balancing the enhancement of reasoning performance with cost control. Its recent deployment in GPT-5 (OpenAI, 2025) further demonstrates the great potential of this approach.

Recent studies have increasingly explored model routers in various scenarios. One line of research aims to select one or more models that are most suitable for each task from a knowledge coverage perspective (Feng et al., 2024; Zhang et al., 2025; Dekoninck et al., 2024; Chen et al., 2024c). This approach can be viewed as a form of LLM ensembling, motivated by the observation that different LLMs exhibit complementary strengths in different knowledge domains. While effective for knowledge-intensive tasks such as factual QA, these methods are limited when applied to multi-step complex reasoning (e.g., mathematical derivations), and they seldom explicitly optimize for inference cost efficiency. Another line of work focuses on device-cloud collaboration, where local lightweight

small language models (SLMs) and cloud-based LLMs are coordinated such that simpler tasks are routed to SLMs, while more complex tasks are escalated to LLMs (Chen et al., 2024a; Shao et al., 2025a; Li et al., 2019; Hao et al., 2024). However, operating at the task level often results in overly coarse routing granularity, making accurate routing decisions challenging and introducing additional overhead.

To address these limitations, we revisit the problem of model routing from the perspective of sub-tasks. Even complex reasoning problems often comprise relatively simple sub-tasks, which can be effectively resolved by more computationally efficient small-scale language models (SLMs). If these simpler "thoughts" can be accurately identified and delegated to such SLMs, while reserving the more complex, capability-intensive sub-problems for larger LLMs, the overall cost can be substantially reduced. This hierarchical approach to model utilization aligns naturally with typical data center deployment scenarios, where a diverse set of models with varying capabilities is often available, enabling dynamic allocation based on sub-task requirements.

Nevertheless, implementing such a framework faces two core challenges. First, high-quality task decomposition, splitting the overall problem into coherent, solvable sub-tasks, is non-trivial (Wies et al., 2023; Zhou et al., 2022), as poor decomposition can produce erroneous intermediate steps or inefficient work allocation, undermining both outcomes and efficiency (Zhu et al., 2023; Zheng et al., 2023). Second, determining the difficulty of each sub-task is challenging but critical for assigning the right model; errors may overload smaller models or waste larger ones, reducing inference efficiency, accuracy, and the potential cost savings of this approach.

To overcome these challenges, we propose **R2-Reasoner**, a framework that leverages a *Reinforced Model Router* to efficiently scale LLM reasoning. As the core component, the Router operationalizes task decomposition and subtask allocation as two distinct yet interconnected LLMs: the Task Decomposer generates a structured sequence of sub-tasks from a complex input, while the Subtask Allocator assigns each subtask to the most suitable model, ranging from lightweight SLMs to powerful LLMs, based on estimated difficulty. By explicitly separating decomposition and allocation, R2-Reasoner enables fine-grained, scalable collaboration across heterogeneous models, optimizing both reasoning accuracy and computational efficiency.

To fully unlock the potential of the *Model Router*, we develop a staged reinforcement learning pipeline that progressively refines its decision-making capability. We decouple the joint training of the Decomposer and Allocator, two core LLMs, into an alternating iterative process, avoiding the non-differentiability and gradient blockage in end-to-end updates across multiple LLMs. This approach combines supervised fine-tuning on task-specific data with Group Relative Policy Optimization (GRPO) in a multi-stage pipeline, enabling stable and coordinated policy improvement through self-supervised feedback. The framework requires no additional human annotation and enhances adaptability in dynamic real-world scenarios.

Extensive evaluations across 6 benchmarks validate the efficacy of our framework. The results demonstrate a substantial reduction in inference costs, achieving an 84.46% decrease in API expenses while maintaining reasoning performance competitive with strong baseline methods and even improving average accuracy by 3.73%. Additionally, we conduct further experiments to demonstrate that R2-Reasoner exhibits strong generalization, capable of directly adapting to previously unseen models. Moreover, our framework supports a flexible and controllable trade-off between accuracy and inference cost, enabling practical deployment across diverse budget scenarios. In summary, our key contributions are:

- We propose **R2-Reasoner**, a novel framework centered around a *Reinforced Model Router* designed to efficiently scale LLM reasoning at test-time. This framework facilitates fine-grained, collaborative reasoning by decomposing complex tasks and strategically allocating subtasks across a diverse pool of heterogeneous models.

- We introduce a staged training pipeline to optimize the *Model Router*. This iterative training strategy not only enables the router to iteratively refine its performance but also circumvents the non-differentiability that arises in end-to-end gradient propagation between two LLMs.

- We conduct extensive experiments on six complex reasoning benchmarks, demonstrating that R2-Reasoner can substantially reduce reasoning costs while maintaining high reasoning accuracy, thereby paving the way for more scalable test-time scaling.

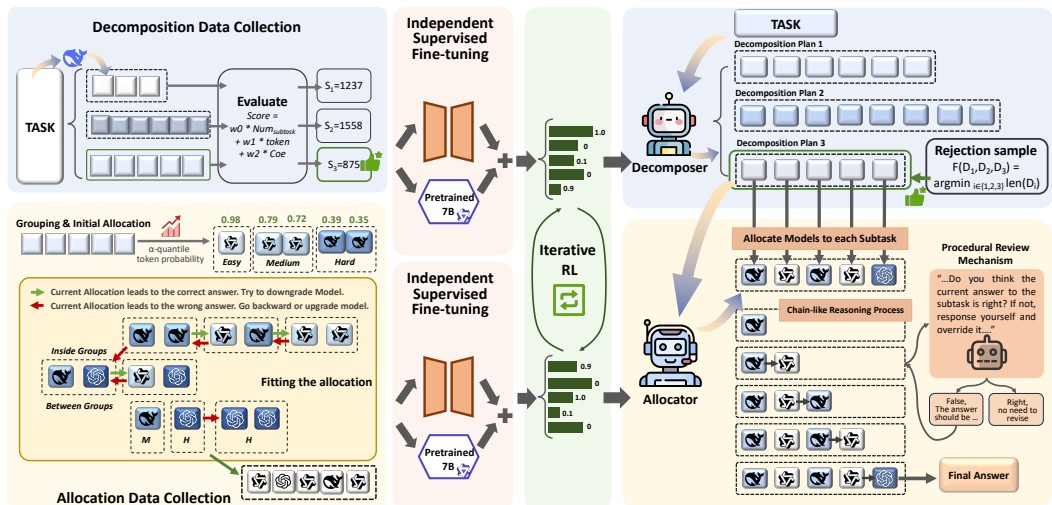

Figure 1: Overview of Our R2-Reasoner Framework

## 2 RELATED WORKS

### 2.1 TASK DECOMPOSITION AND MULTI-STEP REASONING

The chain-of-thought (CoT) prompting technique (Wei et al., 2022) has emerged as a key method for enhancing LLM reasoning, enabling step-by-step inference without additional training. Building on this idea, more advanced paradigms such as tree-of-thought (ToT) (Yao et al., 2023) generalize reasoning into structured sequences of intermediate "thoughts." Leveraging this notion, task decomposition methods and process reward models (Lightman et al., 2023) have been proposed to guide or supervise individual reasoning steps. Together, these approaches illustrate an emerging paradigm that scales reasoning through both structural decomposition and increased compute (Snell et al., 2024).

### 2.2 COLLABORATIVE REASONING AMONG LLMS

Recent research has explored several strategies for enabling collaborative reasoning among multiple language models, each with distinct trade-offs. Model partitioning (Li et al., 2019; Cai et al., 2024; Zhang et al., 2024) distributes a single LLM across nodes, but suffers from high communication overhead and limited robustness. Simple referral (Chen et al., 2024a) routes easy queries to small models and harder ones to stronger LLMs, though performance depends on accurately assessing query difficulty. Token correction (Hao et al., 2024) lets an SLM draft outputs while an LLM revises suboptimal tokens, improving quality but incurring extra decoding costs. Despite these advances, existing methods remain constrained by coordination efficiency, accuracy, and scalability, underscoring the need for more adaptive collaboration frameworks.

## 3 PRELIMINARIES

**Problem Definition**

We consider a highly general scenario of a large-scale LLM platform, where numerous models are deployed locally on the platform, and some are hosted in the cloud. The goal of the platform is to make comprehensive use of these LLMs to provide users with high-quality inference services at the lowest possible cost. Denote the local deployed SLMs as $\mathcal{M}_{\mathcal{E}} = \{\mathcal{M}_{\mathcal{E}_1}, \mathcal{M}_{\mathcal{E}_2}, \ldots, \mathcal{M}_{\mathcal{E}_n}\}$, and the cloud-based LLMs as $\mathcal{M}_{\mathcal{C}} = \{\mathcal{M}_{\mathcal{C}_1}, \mathcal{M}_{\mathcal{C}_2}, \ldots, \mathcal{M}_{\mathcal{C}_q}\}$. The user's original query is restricted to the edge model for task decomposition and allocation, while the resulting sub-tasks can be resolved by either $\mathcal{M}_{\mathcal{E}}$ or $\mathcal{M}_{\mathcal{C}}$. The entire set of reasoning tasks is represented as $\mathcal{T} = \{T_1, T_2, \ldots, T_n\}$. Let the reasoning accuracy over the entire task set be denoted as $Acc$, with the API cost represented by $C_{\text{Api}}$.

For each task $T$, denote the decomposition process as: $T \rightarrow \{t^1, t^2, \ldots, t^k\}$. Based on the decomposed subtasks $t^i$, the model allocation scheme can be denoted as: $M : t^i \mapsto \{\mathcal{M}_{\mathcal{E}}, \mathcal{M}_{\mathcal{C}}\}$, which prioritizes assigning simple subtasks to on-device SLMs, while invoking the cloud-based LLM for

handling complex subtasks. The goal of our optimization is to minimize the discrepancy between the model's allocation scheme $M$ and the optimal scheme $M^*$: $\min |M - M^*|$. The optimal scheme $M^*$ is derived through a search strategy that maximizes SLM usage while maintaining accuracy. During the optimization process, as the allocation scheme gradually approaches the optimal solution, the API cost $C_{\text{Api}}$ decreases, while $Acc$ remains well-maintained.

## 4 Methodology

The R2-Reasoner framework is centered around a **Model Router**, which consists of two primary modules: a **Task Decomposer** ($\mathcal{M}_{\text{decomp}}$) and a **Subtask Allocator** ($\mathcal{M}_{\text{alloc}}$). The Task Decomposer is engineered to break down complex input tasks $T$ into more manageable, well-structured, and logically ordered subtasks $\{t^1, t^2, \ldots, t^k\}$. Following this, the Subtask Allocator strategically assigns each subtask $t^i$ to the most suitable model from a heterogeneous pool ($\mathcal{M}_{pool} = \mathcal{M}_{\mathcal{E}} \cup \mathcal{M}_{\mathcal{C}}$, comprising models of diverse capabilities). This allocation process is driven by the estimated difficulty of each subtask, aiming to strike an optimal balance between reasoning fidelity and computational resource expenditure. The design and training of these interconnected components are detailed below.

### 4.1 Generating Coherent Subtask Sequences via Task Decomposer

The **Task Decomposer** ($\mathcal{M}_{\text{decomp}}$) serves as the first stage of the Model Router, responsible for transforming a complex task $T$ into a sequence of logically connected subtasks $\{t^1, t^2, \ldots, t^k\}$. The quality of this decomposition is crucial: redundant or incoherent breakdowns can cause error propagation, while clear and concise subtasks provide a strong foundation for subsequent allocation.

To supervise training, we construct a decomposition dataset $\mathcal{D}_{\text{decomp}}$ using a rejection sampling strategy. For each task, multiple candidate decompositions are generated and then evaluated along three dimensions: **Conciseness**, assessed by the number of subtasks to avoid both excessive fragmentation and overly coarse splits. **Practicality**, estimated by the total token cost of solving all subtasks with a baseline model. **Coherence**, measuring the logical continuity between adjacent subtasks, with fewer breaks indicating higher quality.

These criteria are linearly combined into a weighted score, where lower values correspond to higher-quality decompositions. A binary correctness signal $C(d) \in \{0, 1\}$ is further incorporated to ensure that the selected decomposition can solve the original task. When possible, only candidates with $C(d) = 1$ are retained, and among them the one with the best score is chosen. This guarantees that $\mathcal{D}_{\text{decomp}}$ contains decompositions that are concise, coherent, and practical while remaining effective for solving the task. The resulting pairs $(T, d^*)$ are then used to fine-tune $\mathcal{M}_{\text{decomp}}$. More details and formulas are provided in the Appendix B.1.

### 4.2 Strategic Model Assignment for Collaboration via Subtask Allocator

Once $\mathcal{M}_{\text{decomp}}$ produces a subtask sequence, the **Subtask Allocator** ($\mathcal{M}_{\text{alloc}}$) determines how to distribute these subtasks across the heterogeneous model pool $\mathcal{M}_{pool}$. Formally, for each subtask $t^i$, it selects a model $M_j \in \mathcal{M}_{pool}$, yielding an assignment $M_A : t^i \mapsto M_j$. To enable $\mathcal{M}_{\text{alloc}}$ to learn efficient assignment policies, we construct a high-quality dataset $\mathcal{D}_{\text{alloc}}$ of model allocation schemes. Rather than relying on hand-crafted heuristics, we employ a systematic search procedure over the vast space of possible assignments, seeking schemes that minimize resource consumption while maintaining perfect accuracy. The resulting allocation pairs $(\{t^i\}, M_A^*)$ serve as supervision signals for training $\mathcal{M}_{\text{alloc}}$ to imitate these cost-effective strategies.

However, exhaustive search over all allocations would be prohibitively expensive in both time and cost. We therefore design a **Grouped Search Strategy** to approximate optimal assignments efficiently. The process begins by estimating the difficulty of each subtask $t^i$ using the predictive confidence of a baseline model: if the maximum token probability exceeds a threshold $\tau_{\text{easy}}$, the subtask is labeled as *easy*; if it falls below $\tau_{\text{hard}}$, it is labeled as *hard*; otherwise, it is labeled as *medium*. In parallel, the model pool $\mathcal{M}_{pool}$ is partitioned into three capability groups: small language models (SLMs), medium language models (MLMs), and large language models (LLMs). Each difficulty level is paired with the corresponding capability group (easy→SLM, medium→MLM, hard→LLM).

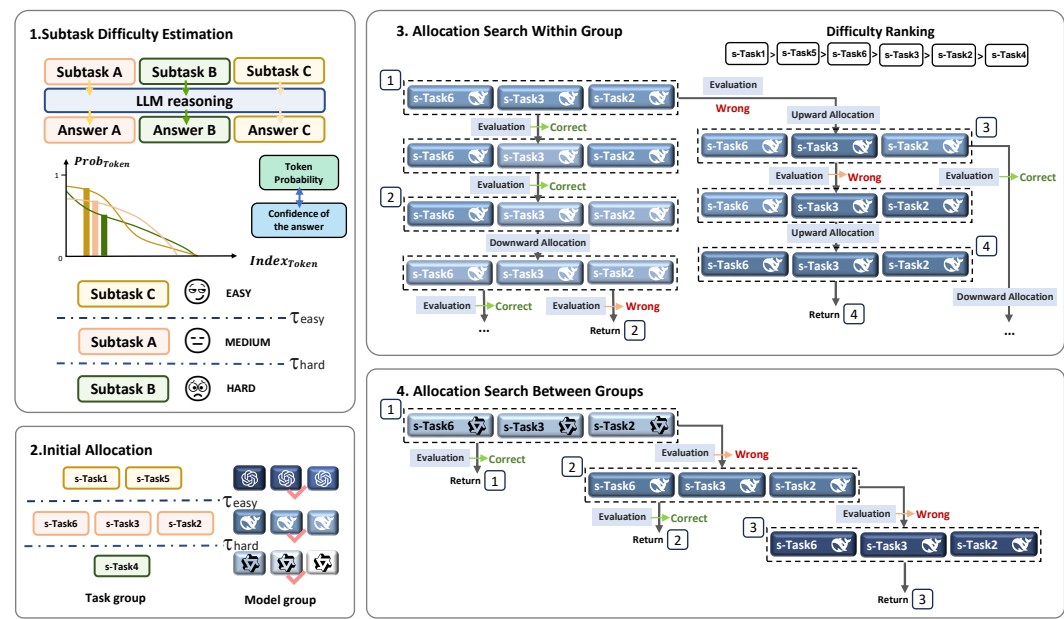

Figure 2: Overview of Our Grouped Search Strategy for Optimal Allocation Scheme

Based on this categorization, an **initial allocation** $M_A^{(0)}$ is obtained by assigning each subtask to the medium-capacity model within its corresponding group. This serves as the starting point for iterative refinement: if the current allocation already achieves correctness ($Acc = 1$), the allocator attempts to replace some models with cheaper ones to reduce cost; if correctness fails, subtasks are escalated to stronger models within the same group, and only if necessary, across groups. The search is bounded by a maximum number of iterations ($N_{\text{iter\_alloc}} \leq 20$), after which the resulting allocation $M_A^*$ is accepted. The collection of such $(\{t^i\}, M_A^*)$ pairs constitutes $\mathcal{D}_{\text{alloc}}$, which is then used to train $\mathcal{M}_{\text{alloc}}$. Details of the search algorithm is shown in Algorithm 1.

This strategy enables $\mathcal{M}_{\text{alloc}}$ to learn fine-grained, capability-aware assignment policies that balance accuracy and efficiency. The detailed formulation of the grouped search procedure is deferred to Appendix B.2.

### 4.3 DUAL-MODULE CO-TRAINING VIA ITERATIVE REINFORCEMENT LEARNING

After the initial SFT of $\mathcal{M}_{\text{decomp}}(\theta_{\text{decomp}})$ and $\mathcal{M}_{\text{alloc}}(\theta_{\text{alloc}})$, We employ a staged RL pipeline to further refine their capabilities and promote synergistic collaboration within the Model Router. In each iteration, one module's parameters are updated while the other remains fixed, allowing targeted improvements based on task success feedback, which also circumvents the non-differentiability and discontinuities arising from cascading two LLMs, thereby stabilizing training. The primary reward signal is a binary indicator based on the final correctness of the task $T$:

$$R_{final}(T, \{t^i\}, M_A) = \begin{cases} 1 & \text{if final answer is correct} \\ 0 & \text{if final answer is incorrect} \end{cases} \quad (1)$$

We adopt Group Relative Policy Optimization (GRPO) as the optimization algorithm for this co-training phase. Training proceeds iteratively for each module:

1. **Updating $\mathcal{M}_{\text{decomp}}(\theta_{\text{decomp}})$**: The decomposer acts as the policy, generating sequences of subtasks $\{t^i\}$ for an input task $T$. The fixed allocator $\mathcal{M}_{\text{alloc}}(\bar{\theta}_{\text{alloc}})$ assigns models to these subtasks, and the final outcome is used to compute $R_{final}$. The reward is propagated back to estimate the advantage $\hat{A}_{i,k}$ for decomposition decisions.

2. **Updating $\mathcal{M}_{\text{alloc}}(\theta_{\text{alloc}})$**: The allocator acts as the policy, generating assignments $M_A(t^k)$ for each subtask $t^k$ provided by the fixed decomposer $\mathcal{M}_{\text{decomp}}(\bar{\theta}_{\text{decomp}})$. The final correctness again determines $R_{final}$, which guides the advantage estimates $\hat{A}_{i,k}$ for allocation choices.

This alternating optimization encourages the two modules to progressively adapt to each other, leading to improved overall reasoning performance. The detailed algorithmic design can be found in Appendix B.3.

## 4.4 END-TO-END REASONING WORKFLOW AT TEST TIME

With the R2-Reasoner's Task Decomposer ($\mathcal{M}_{\text{decomp}}$) and Subtask Allocator ($\mathcal{M}_{\text{alloc}}$) trained through SFT and the iterative RL pipeline, the framework can be deployed for inference. For a user query $Q_{user}$, the workflow is as follows: (1) **Task Decomposition**: The query $Q_{user}$ is first processed by the fine-tuned Task Decomposer: $\{t^1, \ldots, t^k\} = \mathcal{M}_{\text{decomp}}(Q_{user})$. (2) **Subtask Allocation**: The resulting sequence of subtasks $\{t^1, \ldots, t^k\}$ is then passed to the fine-tuned Subtask Allocator for strategic model assignment: $M_A = \mathcal{M}_{\text{alloc}}(\{t^1, \ldots, t^k\})$, where $M_A(t^i) \in \mathcal{M}_{pool}$ is the model assigned to subtask $t^i$. (3) **Subtask Execution**: Each subtask $t^i$ is executed sequentially by its assigned model $M_A(t^i)$. The output of subtask $t^i$ can serve as input to the subsequent subtask $t^{i+1}$. (4) **Result Integration**: The results from the executed subtasks are sequentially integrated to formulate the final answer $A_{\text{final}}$.

To flexibly adapt to scenarios with different cost budgets, achieve a controllable accuracy–cost trade-off, and enhance reasoning robustness, we introduce an optional **Procedural Review Mechanism** (PRM). Let $\mathcal{M}_{\text{strong}}$ denote a high-capability model (e.g., a frontier LLM from $\mathcal{M}_{pool}$) and $\mathcal{M}_{\text{thresh}}$ a pre-defined threshold model representing a minimum capability level. For each subtask $t^j$, let $r_j$ be the output generated by its initially assigned model $M_A(t^j)$. If $M_A(t^j)$ is below the threshold $\mathcal{M}_{\text{thresh}}$, the output will be verified and potentially refined: $r_j^{\text{final}} = \text{PRM\_Verify}(\mathcal{M}_{\text{strong}}, r_j)$ The PRM_Verify function utilizes $\mathcal{M}_{\text{strong}}$ to assess the correctness of $r_j$. If $r_j$ is deemed incorrect or suboptimal, $\mathcal{M}_{\text{strong}}$ provides a corrected or refined response $r_j'$; otherwise, $r_j^{\text{final}} = r_j$. This $r_j^{\text{final}}$ is then used for all subsequent reasoning steps. This mechanism allows targeted quality control, preserving accuracy while maintaining the cost-efficiency of allocation.

## 5 EXPERIMENTS

### 5.1 EXPERIMENTAL SETUP

**Benchmarks**

We evaluate our framework on six widely-used open-source benchmarks: (1) **P3** (Schuster et al., 2021) for program synthesis, (2) **SCAN** (Lake & Baroni, 2018) for language-driven navigation, (3) **MATH** (Hendrycks et al., 2021) and **CHAMP** (Mao et al., 2024)for solving challenging math problems, and (4) **CSQA** (Talmor et al., 2018) and **MuSiQue** (Trivedi et al., 2022) for commonsense reasoning. For each benchmark, we manually annotate a small set of samples for in-context learning in task decomposition and select another 200 tasks as the test set. Detailed descriptions and dataset statistics are provided in the Appendix C.2.

**Baselines** Considering the scenario of collaborative reasoning, we establish six baselines. **(1) CoT (Wei et al., 2022):** The CoT (Chain of Thought) method asks a single LLM to solve a task by decomposing the original task into a sequence of sub-tasks and answering these sub-tasks sequentially. **(2) ToT (Yao et al., 2023):** The ToT (Tree of Thoughts) method, based on the framework of CoT, prompts multiple answers (N = 2) for each sub-task, and retain the best answer by utilizing a scoring method. It also only deploys one certian LLM. **(3) DataShunt (Chen et al., 2024a):** The Datashunt method dynamically selects between a SLM and a LLM to finish the task. The method first evaluates the difficulty of the given task, and allocate the task to either SLM or LLM to solve utilizing the CoT method. **(4) AutoMix (Aggarwal et al., 2025):** The AutoMix method consists of a few-shot self-verification mechanism conducted by SLM to evaluate the confidence toward an answer from SLM and a router that strategically routes queries to LLM based on the confidence. **(5) DoT (Shao et al., 2025b):** The DoT method decomposes a task into subtasks, builds a dependency graph, and allocates subtasks to SLMs or LLMs using a Plug-and-Play Adapter on SLMs. This framework enables efficient edge-cloud collaborative reasoning. **(6) Router-R1 (Zhang et al., 2025):** The Router-R1 method chooses an language model as the router itself, interweaving thinking process by the router with routing process by the routed models, and integrates every response into the context.

| Model | Program Synthesis | | Language-Driven Navigation | | Math Problem Solving | | | | Commonsense Reasoning | | | |
|---|---|---|---|---|---|---|---|---|---|---|---|---|
| | P3 | | SCAN | | MATH | | CHAMP | | CSQA | | MuSiQue | |
| | *Acc* | $C_{API}$ | *Acc* | $C_{API}$ | *Acc* | $C_{API}$ | *Acc* | $C_{API}$ | *Acc* | $C_{API}$ | *Acc* | $C_{API}$ |
| COT (GPT-4o) | **42%** | 4.45¢ | 68% | 2.75¢ | 51.5% | 5.34¢ | 55.5% | 4.45¢ | 80% | 3.60¢ | 57% | 0.85¢ |
| TOT (GPT-4o) | 38% | 14.55¢ | 52% | 9.82¢ | 63% | 9.97¢ | 57% | 11.65¢ | 82% | 20.50¢ | 59% | 2.45¢ |
| COT (Llama 3-8B) | 5.5% | - | 17% | - | 10% | - | 19% | - | 70% | - | 38% | - |
| TOT (Llama 3-8B) | 5.5% | - | 13% | - | 29.5% | - | 25% | - | 68.5% | - | 31% | - |
| DataShunt | 14% | 2.45¢ | 23.5% | 1.72¢ | 16% | 1.66¢ | 34% | 2.98¢ | 73% | 1.28¢ | 47% | 0.46¢ |
| AutoMix | 14% | 0.04¢ | 43% | 0.12¢ | 44% | 0.03¢ | 44% | 0.34¢ | 66% | 0.001¢ | 51% | 0.0074¢ |
| DoT | 41% | 1.58¢ | 63% | 1.20¢ | 59% | 1.02¢ | **58%** | 0.84¢ | 82% | 0.49¢ | 50% | 0.13¢ |
| Router-R1 | 7% | 0.14¢ | 2% | 0.15¢ | 58% | 0.62¢ | 47% | 9.78¢ | 54% | 0.12¢ | 38% | 0.12¢ |
| **R2-Reasoner** | 38% | 1.16¢ | **75%** | 0.64¢ | **76.5%** | 0.08¢ | **59.5%** | 0.28¢ | **83.5%** | 0.042¢ | 56.5% | 0.029¢ |
| Improvement | ↓9.52% | ↓73.93% | ↑10.29% | ↓76.73% | ↑21.43% | ↓99.18% | ↑2.59% | ↓66.67% | ↑1.83% | ↓91.43% | ↓4.24% | ↓98.82% |

Table 1: Performance of R2-Reasoner and baselines on 6 benchmarks. $\mathbf{C_{API}}$ is averaged expense for each task, where API cost is measured in US dollar cents (¢). "-" appears in experiments where reasoning is conducted solely using local deployed SLMs without invoking the cloud-based LLMs. The highest reasoning accuracy is highlighted in bold. Results of the baseline with the highest *Acc* are underlined which will be used to compute the "Improvement" in the last row.

**Selection and Deployment of LLMs** For candidate LLMs to solve different subtasks, We select Qwen2.5-0.5B-instruct, Qwen2.5-1.5B-instruct, Qwen2.5-3B-instruct, Qwen2.5-7B-instruct, Qwen2.5-14B-instruct, Qwen2.5-32B-instruct, Qwen2.5-72B-instruct (Qwen et al., 2025), DeepSeek-V3 (DeepSeek-AI et al., 2025), gpt-4o (OpenAI, 2024) as the LLM pool. The ability of these LLMs increases following the order above. Among these models, Qwen2.5-0.5B-instruct, Qwen2.5-1.5B-instruct, Qwen2.5-3B-instruct, Qwen2.5-7B-instruct are fee free for being locally deployed, while the other cloud-based LLMs charges, and the price of the these LLMs also increases following the order above. For SFT and RL training on the task decomposer and subtask allocator, we select Qwen2.5-7B-instruct as the base model.

**Evaluation** For evaluation, we set two metrics: *Acc* and $C_{API}$, which represents our two main concerns in LLM reasoning. *Acc* measures the accuracy of our framework and the baselines on four benchmarks. $C_{API}$ measures the average API cost for a single task, calculated in US dollar cents.

## 5.2 MAIN RESULTS

The comparison between our framework and the baselines in six benchmarks are shown in Table 1. We have highlighted in bold the highest accuracy results among the four baseline experiments on each benchmark, while the associated time costs and API costs are underlined. We compute the relative improvement of our results compared to the baseline with the highest accuracy. The experimental results demonstrate that our framework significantly reduces the API cost while retaining a comparable reasoning accuracy. The relative changes in accuracy compared to the highest baseline accuracy are: -9.52%, +10.29%, +21.43%, +2.59%, +1.83%, -4.24%. Even for P3, the decline in accuracy is still acceptable. The boost in accuracy on benchmark like MATH and SCAN validate the potential of our work in enhancing reasoning ability. Meanwhile, our framework achieves a tremendous reduction in API cost compared to the baseline with the highest accuracy, reaching averagely a decline of 84.46%. The accuracy of our framework on benchmarks like MATH and SCAN surpassing the CoT and ToT method shows the potential disadvantage of excessive reasoning. It usually happens in reasoning process conducted by LLMs of large scale, often deviates from the correct and suitable answer for a subtask because it automatically proceed with reflective or divergent thinking. We design several precise and exquisite prompts attempting to avoid the phenomenon.

## 5.3 ABLATION STUDY

To rigorously evaluate the contribution and of each stage, we report the performance metrics (*Acc* and $C_{API}$) of the Task Decomposer after each training stage, as summarized in Table 2. The table compares the base model, the model after supervised fine-tuning (SFT), and the final model after SFT combined with RL.

As observed, the SFT stage improves performance across all benchmarks compared to the base model. Importantly, the addition of the RL stage consistently further enhances both accuracy and cost efficiency on every task. For instance, accuracy increases by 5–8% on most benchmarks, while

| Stages | P3 | | SCAN | | MATH | | CHAMP | | CSQA | | MuSiQue | |
|---|---|---|---|---|---|---|---|---|---|---|---|---|
| | $Acc$ | $C_{API}$ | $Acc$ | $C_{API}$ | $Acc$ | $C_{API}$ | $Acc$ | $C_{API}$ | $Acc$ | $C_{API}$ | $Acc$ | $C_{API}$ |
| base | 23.5% | 0.314¢ | 14% | 0.066¢ | 67% | 0.150¢ | 50% | 0.494¢ | 70.5% | 0.147¢ | 43% | 0.0226¢ |
| w/ SFT | 33% | 2.027¢ | 68% | 0.577¢ | 75.5% | 0.079¢ | 58% | 0.370¢ | 82% | 0.056¢ | 51.5% | 0.0301¢ |
| w/ SFT+RL | 38% | 1.160¢ | 75% | 0.636¢ | 76.5% | 0.080¢ | 59.5% | 0.280¢ | 83.5% | 0.042¢ | 56.5% | 0.0287¢ |

Table 2: Performance ($Acc$ and $C_{API}$) after each training stage.

$C_{API}$ is reduced or maintained at a comparable level. This consistent improvement demonstrates that the RL stage not only reliably enhances task performance but also stabilizes the routing decisions across tasks. Overall, these results strongly validate the effectiveness and robustness of our RL-based multi-stage training process.

## 5.4 GENERALIZATION TO NEWLY UNSEEN LLMs

To evaluate the generalization capability of the proposed R2-Reasoner, we conduct an additional experiment in which several models are replaced with alternatives of comparable capacity, without retraining the framework. Specifically, Qwen2.5-7B is replaced with GLM-4-9B-Chat (GLM et al., 2024), and DeepSeek-V3 with Kimi-K2-Instruct (Team et al., 2025). The results are summarized in Table 3.

| Models | P3 | | SCAN | | MATH | | CHAMP | | CSQA | | MuSiQue | |
|---|---|---|---|---|---|---|---|---|---|---|---|---|
| | $Acc$ | $C_{API}$ | $Acc$ | $C_{API}$ | $Acc$ | $C_{API}$ | $Acc$ | $C_{API}$ | $Acc$ | $C_{API}$ | $Acc$ | $C_{API}$ |
| Initial Pool | 38% | 1.160¢ | 75% | 0.636¢ | 76.5% | 0.080¢ | 59.5% | 0.280¢ | 83.5% | 0.042¢ | 56.5% | 0.0287¢ |
| Modified Pool | 33.5% | 1.278¢ | 75% | 0.656¢ | 75% | 0.105¢ | 51.5% | 0.310¢ | 81.5% | 0.060¢ | 51.5% | 0.0438¢ |

Table 3: Experimental results of generalization capability of R2-Reasoner to new LLMs

As observed, the performance of our framework remains largely stable on SCAN, MATH and CSQA. Accuracy decreases by 11.8% on P3, 13% on CHAMP and 9% on MuSiQue, which can be attributed to differences in the reasoning capabilities of the replaced models. Meanwhile, $C_{API}$ increases due to the higher API costs associated with the new models. Overall, these results indicate that the framework exhibits robust generalization to previously unseen LLMs. Importantly, the R2-Reasoner does not rely on any particular model; as long as the relative ordering of model capabilities is preserved, the router can maintain stable and reliable performance across different model pools.

## 5.5 TRADE-OFF BETWEEN REASONING COST AND ACCURACY

Our framework supports a flexible trade-off between accuracy and cost, enabling adaptation to different budget scenarios. By adjusting the routing threshold within our R2-Reasoner, we can dynamically balance performance and expenditure. As shown in Figure 3, when compared against the DoT and DataShunt baselines on the MATH and SCAN benchmarks, our method establishes a new Pareto frontier. The results clearly show that R2-Reasoner consistently achieves significantly higher accuracy for a given cost budget, or conversely, reaches a target accuracy at a substantially lower cost than both competing methods.

This remarkable efficiency is quantitatively demonstrated across both datasets. On the MATH benchmark, R2-Reasoner achieves over 70% accuracy for less than 0.08 cents, while the stronger baseline, DoT, requires approximately 6 cents to reach similar performance—a cost reduction of more than 75×. This advantage holds on the SCAN dataset, where our method reaches 60% accuracy for about 0.4 cents, a task that costs the DoT baseline approximately 5 cents. These results empirically prove that our routing mechanism enables highly effective and budget-aware reasoning, offering practical adaptability for diverse real-world deployment scenarios with varying budget constraints.

## 5.6 INFERENCE TIME COMPARISON ACROSS LLM ROUTERS

We conducted additional experiments under a consistent network environment to evaluate the end-to-end reasoning latency of our framework against several baseline methods. Each experiment was performed independently under identical conditions. All API calls were made sequentially in a single

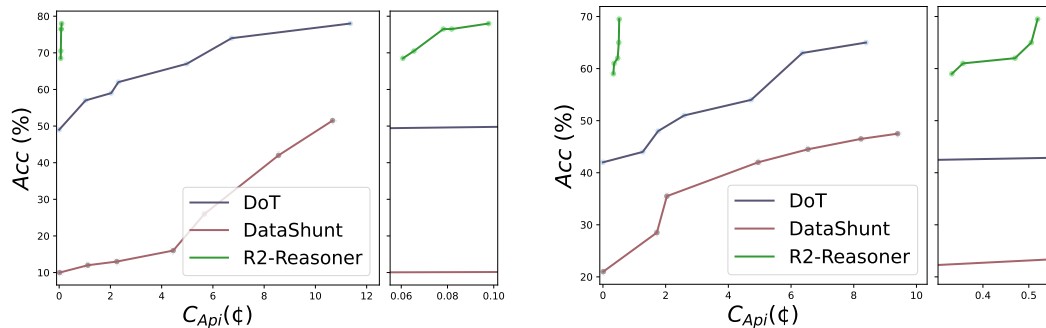

Figure 3: Acc-Cost trade-off curves on MATH (left) and SCAN (right). A magnified inset is provided to the right of the original sub-figure to more precisely illustrate the Pareto frontier of our method.

thread to eliminate concurrency-related interference and ensure that external factors did not distort the latency measurements. The reported results represent the average latency across all tasks in the benchmark, computed after completing full inference runs for every task. In each bar plot, the bar with the darkest color corresponds to our proposed method. The summarized results are presented in Figure 4.

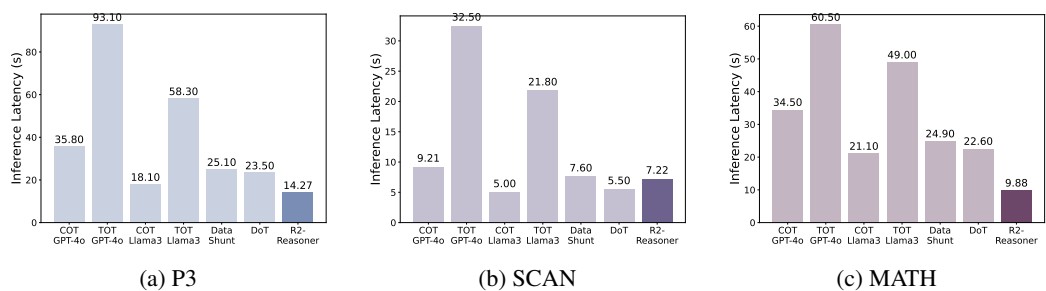

Figure 4: Inference latency comparison of different methods across three benchmarks.

The inference latency results demonstrate significant differences among the evaluated routing methods across the four benchmark tasks. Notably, R2-Reasoner consistently achieves the lowest or near-lowest latency in most cases. For instance, on P3, R2-Reasoner completes inference in 14.27 seconds, substantially faster than CoT and ToT configurations with both GPT-4o and LLaMA 3-8B models, which require between 18.1 and 93.1 seconds. Similar trends are observed on MATH and CSQA, where R2-Reasoner reduces inference time by more than 50% compared to the heaviest baselines (ToT).

On SCAN, R2-Reasoner incurs a slightly higher latency than CoT (LLaMA 3-8B), but it still remains considerably faster than the majority of other methods, including all GPT-4o-based baselines. This performance advantage can be attributed to the framework's adaptive routing strategy, which prioritizes lightweight models for simpler instances and selectively invokes higher-capacity models only when necessary. As a result, R2-Reasoner achieves both time efficiency and cost efficiency, without compromising task performance. Overall, these results highlight the framework's capability to perform fast and scalable reasoning across diverse benchmarks, demonstrating clear practical advantages over existing LLM routing methods.

## 6 CONCLUSION

In this work, we presented **R2-Reasoner**, a novel framework leveraging a reinforced Model Router to efficiently scale large language model reasoning by decomposing complex tasks and allocating subtasks to heterogeneous models. Our staged training pipeline, combining supervised fine-tuning with iterative reinforcement learning, enables adaptive, cost-effective collaboration among models. Looking forward, R2-Reasoner offers promising potential for real-world applications requiring scalable, resource-aware multi-model reasoning, such as complex decision-making systems and cloud computing platform.

## REPRODUCIBILITY STATEMENT

To ensure the reproducibility of our work, we provide all necessary resources and code used in this paper. All benchmarks and models employed are fully open-source or publicly accessible, and no privacy or copyright concerns are involved. All datasets and models are cited in Section 5.1.

Our project code, including the implementation of the R2-Reasoner framework, training scripts, and evaluation pipelines, is publicly available via the following anonymous link: `https://anonymous.4open.science/r/R2_Reasoner`.

Additionally, the main paper, appendix C.2, C.1, and supplementary materials include detailed descriptions of the experimental setup, hyperparameters, and evaluation protocols. Together with the provided code, these materials allow other researchers to fully reproduce the results reported in this work.

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

# A SUPPLEMENTARY EXPERIMENT RESULTS

## A.1 PERFORMANCE IMPROVEMENT OF TASK DECOMPOSER

To evaluate how the two stages of SFT and RL training have improved the task decomposer, we test 100 tasks on four benchmark (one from each category) and report results using two global metrics: $C_d$ and the comprehensive *Score* (defined in Section 4.1). The $C_d$ is calculated as the accuracy of the final answer obtained by allocating all subtasks generated from the current checkpoint to Llama3-8B, while the *Score* is computed following Equation 3. The comparison between the base model and our trained checkpoints is shown in Table 4. On average, SFT and RL jointly yield a 27% increase in $C_d$ and a 6% reduction in *Score*. Across all benchmarks, SFT provides consistent improvements, while RL exhibits mild instability but still contributes overall gains. We attribute this instability to potential insufficiencies in the reward function design.

| Model | P3 | | SCAN | | MATH | | CSQA | |
|---|---|---|---|---|---|---|---|---|
| | $C_d$ | Score | $C_d$ | Score | $C_d$ | Score | $C_d$ | Score |
| base | 0.06 | 2200.51 | 0.38 | 1600.19 | 0.28 | 1311.71 | 0.69 | 1171.53 |
| w/ SFT | 0.10 | 1848.40 | 0.46 | 1557.43 | 0.31 | 1265.60 | 0.75 | 1161.46 |
| w/ SFT+RL | 0.10 | 1788.41 | 0.45 | 1508.50 | 0.34 | 1234.63 | 0.72 | 1201.73 |

Table 4: Performance improvement achieved of the Task Decomposer after multi-stage training.

Beyond these global metrics, we further analyze decomposition quality on three finer-grained dimensions: *Conciseness*, *Practicality*, and *Coherence*. These dimensions are operationalized as follows: *Conciseness*: measured by the number of subtasks generated. *Practicality*: measured by the token cost required for reasoning. *Coherence*: measured by the proportion of logically incoherent subtask pairs (as described in Section 4.1).

| Benchmark | Conciseness | Practicality | Coherence |
|---|---|---|---|
| SCAN | 3.00→2.9263 | 2197.04→2208.89 | 0.1459→0.1367 |
| MATH | 7.54→4.36 | 848.03→939.45 | 0.0364→0.0116 |

Table 5: Evaluation of decomposition quality before and after the training pipeline.

Fewer subtasks are generally preferred, as they directly reduce API cost and latency. An excessive number of subtasks may cause redundancy and confusion. Token consumption is ideally lower, since concise answers are desirable, though moderately longer reasoning chains may yield more thorough inference. For coherence, a smaller value is better, indicating stronger logical consistency among subtasks. As shown in Table 5, our multi-stage training significantly improves decomposition quality across these dimensions, further validating the effectiveness of our approach.

## A.2 PERFORMANCE IMPROVEMENT OF SUBTASK ALLOCATOR

To measure how the 2 stages of SFT and RL training have improved the ability of subtask allocator, we test 100 tasks on each benchmark and set 2 metrics for evaluation: *Acc* and *MAE*. The *Acc* metric measures how many allocation samples are correct according to the labels in our allocation dataset. The *MAE* metrics is based on the LLM pool listed below: Qwen2.5-0.5B-instruct, Qwen2.5-1.5B-instruct, Qwen2.5-3B-instruct, Qwen2.5-7B-instruct, Qwen2.5-14B-instruct, Qwen2.5-32B-instruct, Qwen2.5-72B-instruct, DeepSeek-V3, gpt-4o. Starting from Qwen2.5-0.5B-instruct as model 0, we sequentially assign model indices from 0 to 8, making the size of the number align with the scale of the LLMs. We calculate the MAE between the prediction LLM ID and the label LLM ID. The MAE metric indicates the distance on the LLM map, providing a supplementary sign showing that even if the prediction is wrong, how close it is to the labelled correct answer. The comparison of the base model and our training checkpoint are shown in Table 6. In overall the SFT and RL method have achieved on average 121.29% increase on accuracy and 24.08% decrease on MAE. On all benchmarks, the SFT method shows significant improvement in both metrics. RL method is also slightly unstable but still further achieve an overall improvement on the base of SFT method. The

insuffiency of RL method's effect may be because some inevitable reward hacking during the RL process.

| Model | P3 | | SCAN | | MATH | | CSQA | |
|---|---|---|---|---|---|---|---|---|
| | Acc | MAE | Acc | MAE | Acc | MAE | Acc | MAE |
| base | 0.0923 | 3.0763 | 0.1138 | 3.1041 | 0.1016 | 2.5355 | 0.1773 | 2.5638 |
| w/ SFT | 0.2197 | 2.7762 | 0.2067 | 1.9107 | 0.2362 | 1.8685 | 0.3274 | 1.9419 |
| w/ SFT+RL | 0.2187 | 2.7862 | 0.2606 | 1.9361 | 0.2410 | 1.8603 | 0.3227 | 1.9834 |

Table 6: Performance improvement achieved of the Subtask Allocator after multi-stage training.

### A.3 RL Training Reduces Dependence on SFT Data

To further examine the effectiveness of the RL stage, we conducted an additional experiment on the MATH dataset by deliberately reducing the amount of supervised fine-tuning (SFT) data. Specifically, the SFT training set was reduced by 50%, while the number of RL training epochs was doubled.

Under this setting, the model's accuracy initially dropped by 23% immediately after SFT due to the reduced amount of annotated data. However, after applying RL, not only was this performance degradation fully recovered, but the reasoning accuracy was further improved by an additional 1.5% compared to the original full-data SFT baseline.

This result highlights a key advantage of our RL process: beyond improving reasoning ability, it substantially reduces dependence on large quantities of annotated data. In practice, this suggests that RL can serve as a scalable alternative when labeled resources are limited, making our approach more data-efficient and broadly applicable.

### A.4 Exploring Model Ensembling

Our proposed framework supports flexible extensions and adaptations to different scenarios. As an illustrative case, we evaluate its capability on the task of *model ensembling*. Model ensembling is a widely used strategy that combines the outputs of multiple models in order to improve robustness and potentially enhance accuracy. To enable ensembling within our framework, we design a voting-based mechanism: multiple models are assigned to the same subtask in parallel, and the final answer is determined via majority voting. This mechanism serves as a drop-in replacement for the single-model assignment in our allocator.

As shown in Table 7, ensembling improves the accuracy on MATH, but it does not provide consistent advantages on other benchmarks. Upon further analysis, we suspect that introducing additional models may also introduce misleading signals, which can interfere with the reasoning process of the framework. This experiments demonstrate that our framework not only supports rerouting after failure, but also generalizes to multi-model allocation for a single subtask when ensemble behavior is desired.

### A.5 Clarifications on Potential Bias in Constructed Training Dataset

In the construction of our task decomposition dataset, we applied uniform evaluation metrics across different models, such as token counts, to ensure comparability. To mitigate potential biases arising from inherent differences in how models generate responses (e.g., varying token length distributions), we conducted additional experiments. Specifically, we measured the average token consumption of

| Model | P3 | | SCAN | | MATH | | CSQA | |
|---|---|---|---|---|---|---|---|---|
| | Acc | $C_{API}$ | Acc | $C_{API}$ | Acc | $C_{API}$ | Acc | $C_{API}$ |
| R2-Reasoner | 38% | 1.160¢ | 75% | 0.640¢ | 76.5% | 0.080¢ | 83.5% | 0.042¢ |
| R2-Reasoner w/ Ensembling | 38% | 2.577¢ | 54.5% | 0.934¢ | 83% | 0.222¢ | 81.5% | 0.105¢ |

Table 7: Performance comparison of R2-Reasoner with and without model ensembling.

each model on two benchmark datasets and computed the mean and standard deviation across models. The results are summarized in Table 8.

| Model Name | CSQA Token Num | MATH Token Num |
|---|---|---|
| Qwen2.5-0.5B | 97.22 | 192.15 |
| Qwen2.5-1.5B | 82.75 | 105.37 |
| Qwen2.5-3B | 69.72 | 61.17 |
| Qwen2.5-7B | 58.99 | 48.72 |
| Llama3-8B | 58.34 | 64.42 |
| Qwen2.5-14B | 49.03 | 78.65 |
| Qwen2.5-32B | 52.98 | 116.10 |
| Qwen2.5-72B | 48.51 | 95.01 |
| DeepSeek-V3 | 63.91 | 60.26 |
| GPT-4o | 57.28 | 71.10 |
| Mean | 63.87 | 89.29 |
| Std. Dev. | 15.55 | 42.07 |

Table 8: Average token consumption on CSQA and MATH.

The results show no substantial or systematic variation in token consumption across models, indicating that differences in token usage are not a major source of variability. Instead, the primary source of cost variation lies in the per-token inference cost, which is positively correlated with the model's parameter scale.

## B  FURTHER SUPPLEMENTS TO THE METHODS AND FORMULAS

### B.1  DETAILED FORMULATION OF THE TASK DECOMPOSER

Here, we provide a detailed formulation of the dataset construction process for the Task Decomposer (4.1). The Task Decomposer, denoted as $\mathcal{M}_{\text{decomp}}$, is responsible for transforming a complex input task $T$ into a sequence of clearly defined and logically connected subtasks: $T \xrightarrow{\mathcal{M}_{\text{decomp}}} \{t^1, t^2, \ldots, t^k\}$, where $k$ is the number of subtasks. To systematically evaluate and select high-quality decompositions, we define three complementary metrics. **Conciseness** measures the number of subtasks $k$, balancing between over-fragmentation and overly coarse decomposition. **Practicality** estimates the computational cost by summing the token usage of all subtasks under a baseline evaluation model $\mathcal{M}_{\text{eval}}$:

$$\text{Practicality}(d) = \sum_{i=1}^{k} \text{Tokens}(t^i, \mathcal{M}_{\text{eval}}). \tag{2}$$

**Coherence** evaluates the logical flow by counting adjacent subtask pairs that lack meaningful connection, denoted as $\text{Coe}_{\text{pair}}(d)$. Lower values indicate better continuity.

These metrics are combined into an overall score for a candidate decomposition $d = \{t^i\}_{i=1}^{k}$:

$$\text{Score}(d) = w_c \cdot k + w_p \cdot \sum_{i=1}^{k} \text{Tokens}(t^i, \mathcal{M}_{\text{eval}}) + w_d \cdot \text{Coe}_{\text{pair}}(d), \tag{3}$$

where $w_c, w_p, w_d > 0$ are weighting coefficients. Lower scores correspond to higher-quality decompositions.

Additionally, a binary correctness signal $C(d) \in \{0, 1\}$ is determined by attempting to solve the original task using decomposition $d$ with the evaluation model $\mathcal{M}_{\text{eval}}$. For each task $T$, we generate a set of candidate decompositions $\mathcal{S}_T = \{d_1, d_2, \ldots, d_m\}$ and select the decomposition $d^*$ that minimizes the score while satisfying correctness if possible:

$$d^* = \begin{cases} \arg\min_{d \in \mathcal{S}_T, C(d)=1} \text{Score}(d) & \text{if any } C(d) = 1, \\ \arg\min_{d \in \mathcal{S}_T} \text{Score}(d) & \text{otherwise.} \end{cases} \tag{4}$$

The collection of all $(T, d^*)$ pairs forms the decomposition dataset $\mathcal{D}_{\text{decomp}}$.

Finally, the Task Decomposer is fine-tuned on $\mathcal{D}_{\text{decomp}}$ using a standard cross-entropy loss:

$$\mathcal{L}_{\text{decomp}} = - \sum_{(T,d^*) \in \mathcal{D}_{\text{decomp}}} \sum_i \log P_{\theta_{\text{decomp}}}(d_i^* \mid T), \tag{5}$$

where $d_i^*$ denotes the $i$-th subtask in the target decomposition. This training ensures that $\mathcal{M}_{\text{decomp}}$ consistently generates concise, practical, and coherent subtask sequences suitable for efficient reasoning.

## B.2 GROUPED SEARCH STRATEGY FOR ALLOCATOR TRAINING

Here, we provide the full details of the grouped search algorithm used to construct the allocation dataset $\mathcal{D}_{\text{alloc}}$ (4.2).

**Formal Problem.** Given subtasks $\{t^i\}$ from $\mathcal{M}_{\text{decomp}}$ and a model pool $\mathcal{M}_{pool}$, the objective is to find an allocation scheme $M_A^*$ that minimizes resource consumption while ensuring correctness:

$$M_A^* = \arg \min_{M_A} \mathbb{E}[C_{Api}(M_A) + C_{Time}(M_A)] \quad \text{s.t.} \quad Acc(M_A) = 1. \tag{6}$$

**Granularity Expansion.** Each subtask $t^i$ is labeled with a difficulty level based on $\alpha$-quantile token probabilities:

$$G(t^i) = \begin{cases} G_E & p(t^i) \geq \tau_{diff1}, \\ G_M & \tau_{diff2} < p(t^i) < \tau_{diff1}, \\ G_H & p(t^i) \leq \tau_{diff2}. \end{cases} \tag{7}$$

Simultaneously, models are grouped by capability:

$$\mathcal{M}_{pool} = \mathbb{G}_{\mathcal{M}}^{SLM} \cup \mathbb{G}_{\mathcal{M}}^{MLM} \cup \mathbb{G}_{\mathcal{M}}^{LLM}. \tag{8}$$

An initial allocation $M_{A,0}$ maps each subtask to the medium-capacity model within the corresponding group.

**Within-Group Refinement.** For each iteration $j$, the allocation $M_{A,j}$ is updated as:

$$M_{A,j+1}(t^i) = \begin{cases} \text{smaller}(\mathbb{G}_{\mathcal{M}}^X) & \text{if } Acc(M_{A,j}) = 1, \\ \text{larger}(\mathbb{G}_{\mathcal{M}}^X) & \text{if } Acc(M_{A,j}) = 0, \end{cases} \tag{9}$$

where $X = G(t^i)$.

**Cross-Group Adjustment.** If correctness cannot be achieved with within-group adjustments, inter-group changes are made:

$$M_{A,j+1}(t^i) \in \mathbb{G}_{\mathcal{M}}^Y, \quad Y \neq X, \tag{10}$$

subject to available model capacities.

**Termination.** The algorithm halts after at most $N_{\text{iter\_alloc}} \leq 20$ iterations or when $Acc(M_{A,j}) = 1$ with minimal resource usage. The resulting allocations $\{(\{t^i\}, M_A^*)\}$ populate $\mathcal{D}_{\text{alloc}}$.

**Training Objective.** The allocator $\mathcal{M}_{\text{alloc}}$ is trained on $\mathcal{D}_{\text{alloc}}$ via supervised fine-tuning. The loss function is defined as:

$$\mathcal{L}_{\text{alloc}} = - \sum_{(\{t^i\}, M_A^*) \in \mathcal{D}_{\text{alloc}}} \sum_i \log P_{\theta_{\text{alloc}}}(M_A^*(t^i) \mid t^i). \tag{11}$$

The algorithmic workflow of grouped search strategy for allocator training is illustrated in Algorithm 1.

---

**Algorithm 1** Grouped Search Strategy for Optimal Allocation Scheme $M_A^*$

---

**Require:** Subtask sequence $\{t^i\}$; Model pool $\mathcal{M}_{pool}$; Difficulty thresholds $\tau_{diff1}, \tau_{diff2}$; Max iterations $N_{\text{iter\_alloc}} \leq 20$.
**Ensure:** Near-optimal allocation scheme $M_A^*$ that minimizes cost with $Acc = 1$.
 1: **Phase 1: Initialization via Granularity Expansion and $\alpha$-quantile**
 2: **for** each subtask $t^i$ **do**
 3:   $G(t^i) \leftarrow$ ClassifyDifficulty$(t^i, \tau_{diff1}, \tau_{diff2})$       $\triangleright$ Categorize as $G_E, G_M, G_H$
 4: **end for**
 5: Partition $\mathcal{M}_{pool}$ into capability groups $\mathbb{G}_{\mathcal{M}}^{SLM}, \mathbb{G}_{\mathcal{M}}^{MLM}, \mathbb{G}_{\mathcal{M}}^{LLM}$
 6: $M_A \leftarrow M_{A,0}$ by mapping each $G(t^i)$ to medium_model$(\mathbb{G}_{\mathcal{M}}^X)$
 7: Initialize best-found scheme $M_A^* \leftarrow$ null

 8: **Phase 2: Iterative Search and Refinement**
 9: **for** $j = 0$ to $N_{\text{iter\_alloc}} - 1$ **do**
10:   **if** EvaluateAccuracy$(M_A) == 1$ **then**
11:     $M_A^* \leftarrow M_A$              $\triangleright$ Update best-found valid scheme
12:     $M_A \leftarrow$ AdjustDown_WithinGroup$(M_A)$  $\triangleright$ Seek a more resource-efficient solution
13:   **else**               $\triangleright$ Accuracy is 0, need a more powerful scheme
14:     $M_{A,\text{next}} \leftarrow$ AdjustUp_WithinGroup$(M_A)$ $\triangleright$ Try upgrading models within their groups
15:     **if** $M_{A,\text{next}} == M_A$ **then**    $\triangleright$ Within-group adjustments are maxed out
16:       $M_A \leftarrow$ AdjustUp_BetweenGroup$(M_A)$ $\triangleright$ Escalate to inter-group model upgrades
17:     **else**
18:       $M_A \leftarrow M_{A,\text{next}}$
19:     **end if**
20:   **end if**
21: **end for**
22: **return** $M_A^*$

---

### B.3 GRPO Objective for Co-training

For completeness, we provide the full GRPO objective function used in our co-training phase (4.3). The general form is:

$$
\begin{aligned}
\mathcal{J}_{\text{GRPO}}(\theta_{RL}) =&\mathbb{E}_{q \sim P(Q), \{o_i\}_{i=1}^G \sim \pi_{\theta_{\text{old}}}(O|q)} \left[ \frac{1}{G} \sum_{i=1}^{G} \frac{1}{|o_i|} \sum_{k=1}^{|o_i|} \left\{ \min \left[ \frac{\pi_{\theta_{RL}}(o_{i,k}|q, o_{i,<k})}{\pi_{\theta_{\text{old}}}(o_{i,k}|q, o_{i,<k})} \hat{A}_{i,k}, \right.\right.\right. \\
&\left.\left.\left. \text{clip} \left( \frac{\pi_{\theta_{RL}}(o_{i,k}|q, o_{i,<k})}{\pi_{\theta_{\text{old}}}(o_{i,k}|q, o_{i,<k})}, 1-\epsilon, 1+\epsilon \right) \hat{A}_{i,k} \right] - \beta \mathbb{D}_{\text{KL}}[\pi_{\theta_{RL}}(\cdot|q, o_{i,<k}) || \pi_{\text{ref}}(\cdot|q, o_{i,<k})] \right\} \right]
\end{aligned}
\tag{12}
$$

Here: - $\theta_{RL}$ are the parameters being optimized. - $\pi_{\theta_{RL}}$ is the current policy, while $\pi_{\theta_{\text{old}}}$ is the policy used to generate trajectories. - $o_{i,k}$ is the $k$-th action in the $i$-th trajectory given context $q$. - $\hat{A}_{i,k}$ is the estimated advantage for that action. - The first term is a clipped surrogate objective (as in PPO), and the second term penalizes deviation from a reference policy $\pi_{\text{ref}}$, controlled by $\beta$.

This formulation is applied identically when training either $\mathcal{M}_{\text{decomp}}$ or $\mathcal{M}_{\text{alloc}}$, depending on which module is currently being updated.

## C EXPERIMENT DETAILS

### C.1 EXPERIMENTAL ENVIRONMENT AND TRAINING HYPERPARAMETERS

The hardware environment used for our experiments and the specific training hyperparameters are summarized in Table 9.

In addition to the hardware specifications and basic training parameters, we also set hyperparameters during dataset construction. During constructing the dataset for the Task Decomposer, we computed a weighted average over the three dimensions of task decomposition, which involves three

hyperparameters: $w_c$, $w_p$, and $w_d$. These three hyperparameters serve as weights for: (1) the total number of subtasks, (2) the total number of tokens used during inference, and (3) the coherence score, respectively. Empirically, these three components exhibit significantly different value ranges across a wide range of tasks. Specifically, our analysis shows that their average values are approximately 5.87 (number of subtasks), 676.59 (token count), and 0.1541 (coherence score). To ensure the comparability of these components during weighted aggregation, our hyperparameter selection strategy is based on normalizing them to a similar scale. Accordingly, we set $w_c = 100$, $w_p = 1$, and $w_d = 1000$, which balances their contributions in the combined scoring function.

To assess the sensitivity, we also conducted experiments using 10 distinct parameter settings during the data construction process. We found that the dataset quality is generally stable when the weight ratios stay within a reasonable balance (i.e., fluctuating within ±30Among the parameters, the first one $w_c$ plays the dominant role, critically affecting the quality of decomposition and the complexity of subsequent reasoning, while the other two serve auxiliary roles. We plan to include a more thorough sensitivity analysis in the revised manuscript.

| Module | Element | Detail |
|---|---|---|
| System | OS | Ubuntu 20.04.6 LTS |
| | CUDA | 12.4 |
| | Python | 3.12.9 |
| | Pytorch | 2.6.0 |
| | trl | 0.17.0 |
| | accelerate | 1.6.0 |
| | peft | 0.15.1 |
| | flash_attn | 2.7.4.post1 |
| | Device | 2*NVIDIA A100 80G |
| Workflow | API | Siliconflow & Microsoft Azure |
| SFT | Mode | Lora |
| | Batch size | 4, 8 |
| | Number of epochs | 2, 3 |
| | Max token length | 2048 |
| | Lora rank | 32, 64 |
| | Optimzer | AdamW |
| | Learning rate | 0.00002, 0.00003 |
| RL Training | Algorithm | GRPO |
| | Number of Generation | 4 |
| | Batch size | 1 |
| | Global step | 1024 |
| | Max token length | 2048 |
| | Optimzer | AdamW |
| | Learning rate | 0.0001, 0.00015 |

Table 9: **Detailed Experimental Settings**

## C.2 DETAILS OF THE BENCHMARKS

We verify the effectiveness of our framework upon six open-source benchmarks. These benchmarks target four distinct aspects of the model's capability, including:

- **(1) Program Synthesis:** We select P3 (Schuster et al., 2021) (Python Programming Puzzle) for evaluation. P3 defines each puzzle by a python program $f$ and evaluate the concerned ability of program synthesis by checking if the candidate input provided by machines could make $f$ return True. By a form comprehensible for both humans and machines, it emphasizes on the ability involved during coding process such as syntax correctness and algorithmic reasoning.

For task decomposer, we randomly choose 1500 puzzles from the original P3 benchmark, and filter out 1085 puzzles with their decomposition results which are valid for SFT training. For subtask allocator, we originally randomly select 2000 puzzles and eventally filter out 1687 puzzles with a total number of over 12000 subtask allocation samples as the candidate dataset. We choose 4000 subtask allocation samples for SFT training on the subtask allocator.

- **(2) Language-Driven Navigation:** We select SCAN (Lake & Baroni, 2018) (Simplified version of the CommAI Navigation) for this evaluation. This benchmark consists of a set of navigation commands with the corresponding action sequences. By instructing machines to convert the navigation commands in natural language into a sequence of actions and comparing the generated sequence sample with the label, it focus on assesing the ability of logical navigation, including traversal, backward reasoning and anomaly detection.

  For task decomposer, we randomly choose 2814 commands out of the original SCAN benchmark, and filter out 1180 commands with their decomposition results for the SFT stage training. For subtask allocator, we originally randomly select 2000 commands and obtain a set of 7708 sub-command allocation samples. We also select 4000 sub-command allocation samples for training the subtask allocator.

- **(3) Solving Math Problems:** We select MATH (Hendrycks et al., 2021) and CHAMP (Mao et al., 2024) for this evaluation. The MATH benchmark consists of 12,500 challenging competition mathematics problems, while the CHAMP benchmark contains 270 diverse high school competition-level math problems. These two mainly involves LLM's conducting computation, memorizing mathematical knowledge and utilizing problem-solving techniques.Solving math problems has been universally acknowledged as a crucial aspect to measure LLM's reasoning ability.

  For task decomposer, we randomly choose 2044 math problems from the original MATH benchmark, and use 1430 problems with their decomposition results for fine-tuning the task decomposer. For subtask allocator, we first select 2000 original math problems from the benchmark. After building our own allocation dataset, we obtain over 7000 sub-problem allocation samples, and choose 4000 sub-problem allocation samples for training and boosting the ability of subtask allocator.

- **(4) Commonsense Reasoning:** We select CSQA (Talmor et al., 2018) (CommonsenseQA) and MuSiQue (Trivedi et al., 2022) for this evaluation. These 2 benchmarks require a broader commonsense knowledge base for LLM. Considering the knowledge base varies as the scale of LLM varies, it is a suitable benchmark to test if different LLMs in our framework could collaborate and compose an integrated knowledge base in commonsense scenario.

  For task decomposer, we randomly select 2273 commonsense queries from the original benchmark, and utilize 1591 out of the queries to finish the training process. For subtask allocator, we obtain an original dataset of 1800 commonsense queries, and obtain over 5500 sub-problem allocation samples, and select nearly 4000 sub-problem allocation samples for SFT training.

# D  DISCUSSIONS

## D.1  ENHANCING LLM REASONING VIA REINFORCEMENT LEARNING

The reasoning process of LLMs can be formulated as a partially observable Markov decision process (POMDP), where context serves as the state, token generation as the action, and the objective is to learn a policy maximizing cumulative reward. Since DeepSeek-R1 (Guo et al., 2025), reinforcement learning has become central to enhancing LLM reasoning (Yu et al., 2025; Zuo et al., 2025; Liu et al., 2025). Group Relative Policy Optimization (GRPO) (Shao et al., 2024) has recently gained widespread attention as a leading RL algorithm: it evaluates batches of outputs, computes relative advantages, and uniformly assigns rewards across tokens. Unlike actor-critic methods relying on value estimators, GRPO avoids estimation bias, instability, and reward hacking (Weng, 2024), achieving more stable and faithful optimization.

## D.2 ALTERNATIVE REWARD DESIGNS

We considered several alternative reward formulations beyond the outcome-based design adopted in our experiments. One natural idea is to provide step-wise (intermediate) rewards. However, in tasks such as 24-point arithmetic or multi-step mathematical reasoning, it is often difficult to accurately assess the quality of intermediate steps without knowledge of the final outcome. This makes step-level reward annotation or computation unreliable in practice.

Another potential direction is to leverage Monte Carlo Tree Search (MCTS) to approximate intermediate rewards. While this strategy can, in principle, provide more informative supervision, it introduces substantial computational overhead and significantly increases the complexity of the data construction pipeline, thereby limiting its scalability.

Inspired by the outcome-driven reward design used in DeepSeek R1, we ultimately adopted a final-outcome-based reward scheme. This approach achieves a good balance between effectiveness and efficiency, while remaining scalable to large-scale training. Our experimental results demonstrate that this reward formulation is both practical and effective for the considered reasoning tasks.

## D.3 BROADER IMPACTS

Our R2-Reasoner framework has the potential to significantly broaden the accessibility and applicability of advanced AI reasoning capabilities. By substantially reducing computational costs and latency associated with complex multi-step reasoning, it can democratize the use of powerful Large Language Models. This could enable smaller organizations, individual researchers, or developers with limited resources to leverage state-of-the-art reasoning techniques that are currently prohibitively expensive. In practical terms, this could spur innovation across various sectors. For instance, in education, it could power more sophisticated and responsive AI tutors capable of breaking down complex problems for students in a cost-effective manner. In scientific research, R2-Reasoner could facilitate more intricate automated hypothesis generation and experimental design by making deep reasoning chains more feasible. For enterprise applications, it could lead to the development of more intelligent and nuanced customer service bots, data analysis tools, or decision support systems that can handle complex queries without incurring excessive operational costs.

Furthermore, the principle of dynamically allocating resources based on sub-task complexity could inspire more sustainable AI practices. By preferentially using smaller, more energy-efficient models for simpler tasks, the overall energy consumption and carbon footprint associated with large-scale AI deployments could be reduced. The framework also encourages the development and utilization of a more diverse ecosystem of language models, fostering innovation in both large and small model architectures. Ultimately, by making sophisticated reasoning more efficient and economical, R2-Reasoner can help unlock new applications and accelerate the integration of AI into various aspects of daily life and industry, fostering more intelligent and adaptive systems.

# E PROMPTS

Below show how we construct our prompts for the four aspects of model's reasoning capability, each aspect taking one benchmark's prompt as an example:

## E.1 PROGRAM SYNTHESIS: P3

Below is the prompt for decomposition data collection on benchmark P3:

```
You will be provided with a Programming Puzzle. The ultimate task is
    to find an input that will make the program return True.
To better accomplish this task, now you need to break the puzzle
    into multiple steps, preferably between 3 and 8 steps.

These steps are organized in a chain-like manner, in which the steps
    are supposed to be solved following a certain order.
```

Meanwhile when writing each broken-down question step on a separate line, the order of the questions should be the order of how to solve these broken-down question steps.

4 examples are as follows:

Program 1:
```
def sat(li: List[int], k=6):
    def prod(nums):
        ans = 1
        for i in nums:
            ans *= i
            return ans
    return min(li) > 1 and len(li) == k and all((1 + prod(li[:i] +
        li[i + 1:])) % li[i] == 0 for i in range(k))
```
Result 1 of decomposed steps:
step 1: Understand the conditions required by the function.
step 2: Choose the length of the list based on k.
step 3: Generate potential elements for the list.
step 4: Calculate the product of all other elements for each element in the list when i = 0 and add 1 to the product.
step 5: Calculate the product of all other elements for each element in the list when i = 1 and add 1 to the product.
step 6: Calculate the product of all other elements for each element in the list when i = 2 and add 1 to the product.
step 7: Calculate the product of all other elements for each element in the list when i = 3 and add 1 to the product.
step 8: Calculate the product of all other elements for each element in the list when i = 4 and add 1 to the product.
step 9: Calculate the product of all other elements for each element in the list when i = 5 and add 1 to the product.
step 10: Verify the divisibility condition for each element.
step 11: Adjust the elements and repeat until a valid list is found.
step 12: Confirm that the list meets all conditions.

Program 2:
```
def sat(indices: List[int], s=\"aeEm%%uIV0imR&xUvQvZf#1z4\"):
    i, j = indices
    return s[i] == s[j] and 0 <= i < j < i + 3
```
Result 2 of decomposed steps:
step 1: Understand there are two conditions need to fulfill for the input indices that i and j in the indices should meet first s[i] == s[j] and 0 <= i < j < i + 3
step 2: Iterate through the string sin a group of 3 characters, s[n] s[n+1] s[n+2]
step 3: Compare the three characters to see if any of two characters are the same.
step 4: If identical strings are found, Count the index of both % in the string s; If no identical characters, move to the consecutive three characters.
step 5: Write the index of two identical characters and yield the final answer of list indices.

Program 3:
```
def sat(path: List[int], weights=[{{1: 20, 2: 1}}, {{2: 2, 3: 5}},
    {{1: 10}}], bound=11):
    return path[0] == 0 and path[-1] == 1 and sum(weights[a][b] for
        a, b in zip(path, path[1:])) <= bound
```
Result 3 of decomposed steps:
step 1: Create a list that fulfill the first contraint to have 0 at index 0.

step 2: Create a list that fulfill the second contraint to have 1 at
   last.
step 3: Given that the sum of weights[a][b] for a, b in zip(path,
   path[1:])) <= bound, we need to find values in the list weights
   that is less than 11.
step 4: First checking if combining step 1 and step 2 to be path
   could be the correct input by calculating sum(weights[a][b] for
   a, b in zip(path, path[1:])) <= bound
step 5: If the previous step is not correct, then think about what
   could be the integer filling between 0 and 1.
step 6: Eliminate the incorrect candidates.
step 7: Fill in the number to the list of integer.
step 8: Verify the if the new list will make the function return
   True.

Program 4:
"name": "LastLetters:3",

```python
def sat(y: List[bool], x=['ryxadec', 'pyfixotibujadyxe',
                          'mopubywewexi witethig 7', ' !',
                          'jethi sed c', 'lotextusavufubynyb',
                          'wuxesafetatextysima pebutextiwafufok',
                          'tuchonip', ' S',
                          'xyvovikofutex pylekazuquekedajota E',
                          'wik xofoxujegerigubo ?',
                          'gipimakude 1', ' O', ' ^',
                          'lakiquuvuhenugu vajyquy P',
                          ' 6', 'fezore', 'vabithin
                              textusichytilejocoke',
                          ' B', 'lasuthasebuvy que &',
                          'mymanuzuzudyc thazufys y', '', ' ?',
                          'gecohywelawu', 'wath']):
    assert len(x) == len(y)
    for s, b in zip(x, y):
        if len(s.split(" ")[-1]) == 1:
            assert b == s[-1].isalpha()
        else:
            assert not b
    return True
```

Result 4 of decomposed steps:
step 1: Determine the length of the list x to ensure y has the same
   length.
step 2: Loop through the list x to check the last word of each
   string.
step 3: Check if the last segment of the string in x (seperated by
   space) have length 1.
step 4: If Step 3 meet, check if that character is alphabetical
   characters.
step 5: If step 4 is true, then the boolean value in list y with
   corresponding index should also be True. If not, False.
step 6: If Step 3 do not meet, the boolean value in list y with
   corresponding index should be False.
step 7: The final result should a list of boolean values.

Now here is the puzzle for you to decompose: {question}
Requirements:
1. The steps broken down should preferably be between 3 to 8 steps.
2. Each step needs to be executable, have a clear meaning, or
   produce a meaningful result.

Answer Format:
The process of solving the problem can be divided into the following
   steps:

```
1. question step 1
2. question step 2
3. question step 3
...
```

Below is the prompt for solving subtasks sequentially on benchmark P3:

```
You will be provided with a Programming Puzzle. Your task is to find
    an input that will make the program return True.
Here is the puzzle:{puzzle}

The data type of your final answer should be {ans_type}.
I have broken this puzzle down into many easier subtasks.
Following the order of the subtasks and solving every subtask in
    sequence lead to finding out the correct input.
I will assign you sub-tasks one by one, and provide the results of
    the previous sub-tasks as a reference for your reasoning.
Please follow the sequence of our subtasks to find the correct input
    .

Now, the first several subtasks are already solved, these subtasks
    listed below following the sequence:{previous_tasks}.

Their answers are listed below, also following the sequence:{
    previous_moves}.

Now you need to solve the subtask: {Step_dict[str(cnt)]}.

Focus exclusively on solving the subtask.
Your answer should be concise and directly address the core
    reasoning process.
Avoid any unnecessary comments, greetings, or expressions of
    enthusiasm. Only provide the essential reasoning process and
    answer.

Please provide the answer to the subtask.
```

Below is the prompt for synthesizing to obtain the final answer on benchmark P3:

```
We are provided with a progamming puzzle. Our task is to find an
    input that will make the program return True.
Here is the puzzle:{puzzle}
The data type of our correct input should be {ans_type}.

I have broken this puzzle down into many easier subtasks.
Following the order of the subtasks and solving every subtask in
    sequence lead to finding out the correct input.
All the subtasks are listed in the order: {previous_tasks}
The answers to all the subtasks are listed in the same order: {
    previous_answs}

We can synthesize the final answer based on all the answers to the
    subtasks.
You must synthesize the final answer strictly based on the provided
    answers to the subtasks, without performing any error correction
     or independent recalculations.
```

```
Even if a subtask answer contains a reasoning mistake or calculation
    error, you must still use it as given.
Do not infer the correct answer based on correct reasoning steps if
    the computed result is incorrect.
Your final synthesis should reflect the exact values and conclusions
    stated in the subtask answers, even if they are incorrect.

The final answer is the input that will make the program return True
    .
Please give the input and just give the answer without any
    additional explanation or clarification.
for example, if the final answer is 3, you are supposed to output 3.
    To output "the answer is 3" is forbidden.
for example, if the final answer is [1,2,3], you are supposed to
    output [1,2,3]. To output "```python [1,2,3]```" is forbidden.
```

Below is the prompt for using task decomposer to decompose the original puzzle into a sequence of subtasks on benchmark P3:

```
You will be provided with a Programming Puzzle. The ultimate task is
    to find an input that will make the program return True.
To better accomplish this task, now you need to break the puzzle
    into multiple steps, preferably between 3 and 8 steps.

These steps are supposed to be solved in a chain-like manner
    following a certain order.
Meanwhile when writing each broken-down step on a separate line, the
    order of the steps should be the order of solving these broken-
    down steps.

1 examples is as follows:
Programming Puzzle:
def sat(indices: List[int], s=\"aeEm%%uIV0imR&xUvQvZf#1z4\"):
    i, j = indices
    return s[i] == s[j] and 0 <= i < j < i + 3
Answer:
The process of solving the programming puzzle can be divided into
    the following steps:
1: Understand there are two conditions need to fulfill for the input
    indices that i and j in the indices should meet first s[i] == s
    [j] and 0 <= i < j < i + 3
2: Iterate through the string sin a group of 3 characters, s[n] s[n
    +1] s[n+2]
3: Compare the three characters to see if any of two characters are
    the same.
4: If identical strings are found, Count the index of both % in the
    string s; If no identical characters, move to the consecutive
    three characters.
5: Write the index of two identical characters and yield the final
    answer of list indices.

Now here is the puzzle for you to decompose: {original_question}
Requirements:
1. The steps broken down should preferably be between 3 to 8 steps.
2. Each step needs to be executable, have a clear meaning, or
    produce a meaningful result.

Answer Format:
The process of solving the programming puzzle can be divided into
    the following steps:
```

```
1. step 1
2. step 2
3. step 3
...
```

Below is the prompt for using subtask allocator to allocate candidate model to a certain subtask on benchmark P3:

```
Now we have a programming puzzle.
We need to find the corrct input that will make the program return
    true.
We decide to break this puzzle into subtasks.
Now we have to solve a subtask, and there are 9 models that can be
    chosen to solve this subtask.
These 9 models are:
qwen2.5-0.5b, qwen2.5-1.5b, qwen2.5-3b, qwen2.5-7b, qwen2.5-14b,
    qwen2.5-32b, qwen2.5-72b, deepseek-V3, gpt-4o.
We list these models in ascending order acoording to their
    capability and the difficulty levels of the subtasks they are
    suitable for.
For example,
qwen2.5-0.5b has the lowest capability thus is suitable for the
    easiest subtask;
gpt-4o has the highest capability thus is suitable for the hardest
    subtask.
Task: choose the most appropriate model from the list above to solve
     the given subtask.
Output only the chosen model's name.

the original puzzle: {original_problem}
the subtask: {subtask}
```

## E.2 LANGUAGE-DRIVEN NAVIGATION: SCAN

Below is the prompt for decomposition data collection on benchmark SCAN:

```
I will give you a piece of natural language command. I need you to
    decompose it to smaller commands.

8 examples are as follows:

Command: "look right after look twice"
Result of decomposition: "look right after look twice" can be solved
     by: "look right", "look twice".

Command: "jump opposite right thrice and walk"
Result of decomposition: "jump opposite right thrice" can be solved
    by: "jump opposite right", "jump opposite right thrice". "walk"
    can be solved by: "walk". So, "jump opposite right thrice and
    walk" can finally be solved by: "jump opposite right", "jump
    opposite right thrice", "walk".

Command: "run left twice and run right"
Result of decomposition: "run left twice" can be solved by: "run
    left", "run left twice". "run right" can be solved by "run right
    ". So, "run left twice and run right" can finally be solved by:
    "run left", "run left twice", "run right".
```

```
Command: "run opposite right"
Result of decomposition: "run opposite right" can finally be solved
    by "run opposite right".

Command: "look opposite right thrice after walk"
Result of decomposition: "look opposite right thrice" can be solved
    by: "look opposite right", "look opposite right thrice". "walk"
    can be solved by "walk". So, "look opposite right thrice after
    walk" can finally be solved by: "look opposite right", "look
    opposite right thrice", "walk".

Command: "jump around right"
Result of decomposition: "jump around right" can be solved by: "jump
    right", "jump around right". So, "jump around right" can
    finally be solved by: "jump right", "jump around right".

Command: "look around right thrice and walk"
Result of decomposition: "look around right thrice" can be solved by
    : "look right", "look around right", "look around right thrice".
     "walk" can be solved by "walk". So, "look around right thrice
    and walk" can finally be solved by: "look right", "look around
    right", "look around right thrice", "walk".

Command: "turn right after run right thrice"
Result of decomposition: "turn right" can be solved by: "turn right
    ". "run right thrice" can be solved by: "run right", "run right
    thrice". So, "turn right after run right thrice" can finally be
    solved by: "turn right", "run right", "run right thrice".

Now the command is {question}, please decompose it into smaller
    commands like the examples.
Answer Format: xxx can be solved by: xxx. xxx can be solved by xxx.
    ... So, xxx can finally be solved by: "subcommand_0", "
    subcommand_1",...
```

Below is the prompt for solving sub-commands sequentially on benchmark SCAN:

```
There is a natural language instruction representing a sequence of
    actions. I need you to translate this sentence from natural
    language into a standardized meta-action sequence."
Here is the instruction:{question}

I have broken this instruction down into some smaller instructions.
    I will assign you sub-instructions one by one, and provide the
    results of the previous sub-instructions as a reference for your
     reasoning.
Please organize your reasoning according to the combination and
    progression of actions.

For your reference, 13 examples for translation together with the
    corresponding explanations are as follows:

Q: "turn left"
A: "turn left" outputs "TURN LEFT".

Q: "turn right"
A: "turn right" outputs "TURN RIGHT".

Q: "jump left"
```

```
A: The output of  jump   l e f t  concatenates: the output of  turn
    l e f t , the output of   jump  . turn   l e f t  outputs
    TURN   L E F T .   jump   outputs   JUMP  . So concatenating
    the output of  turn   l e f t  and the output of   jump
    leads to  TURN   L E F T  +   JUMP  . So the output of  jump
    l e f t  is  TURN   L E F T  +   JUMP  .

Q: "run right"
A: The output of "run right" concatenates: the output of "turn right
   ", the output of "run". "turn right" outputs "TURN RIGHT". "run"
    outputs "RUN". So concatenating the output of "turn right" and
   the output of "run" leads to "TURN RIGHT" + "RUN". So the output
    of "run right" is "TURN RIGHT" + "RUN".

Q: "look twice"
A: The output of "look twice" concatenates: the output of "look",
   the output of "look". "look" outputs "LOOK". So repeating the
   output of "look" two times leads to "LOOK" * 2. So the output of
    "look twice" is "LOOK" * 2.

Q: "run and look twice"
A: The output of "run and look twice" concate+nates: the output of "
   run", the output of "look twice". "run" outputs "RUN". "look
   twice" outputs "LOOK" * 2. So concatenating the output of "run"
   and the output of "look twice" leads to "RUN" + "LOOK" * 2. So
   the output of "run and look twice" is "RUN" + "LOOK" * 2.

Q: "jump right thrice"
A: The output of "jump right thrice" concatenates: the output of "
   jump right", the output of "jump right", the output of "jump
   right". "jump right" outputs "TURN RIGHT" + "JUMP". So repeating
    the output of "jump right" three times leads to ("TURN RIGHT" +
    "JUMP") * 3. So the output of "jump right thrice" is ("TURN
   RIGHT" + "JUMP") * 3.

Q: "walk after run"
A: The output of "walk after run" concatenates: the output of "run",
    the output of "walk". "run" outputs "RUN". "walk" outputs "WALK
   ". So concatenating the output of "run" and the output of "walk"
    leads to "RUN" + "WALK". So the output of "walk after run" is "
   RUN" + "WALK".

Q: "turn opposite left"
A: The output of "turn opposite left" concatenates: the output of "
   turn left", the output of "turn left". "turn left" outputs "TURN
    LEFT". So repeating the output of "turn left" twice leads to "
   TURN LEFT" * 2. So the output of "turn opposite left" is "TURN
   LEFT" * 2.

Q: "turn around left"
A: The output of "turn around left" concatenates: the output of "
   turn left", the output of "turn left", the output of "turn left
   ", the output of "turn left". "turn left" outputs "TURN LEFT".
    So repeating the output of "turn left" four times leads to "TURN
    LEFT" * 4. So the output of "turn around left" is "TURN LEFT" *
    4. Q: "turn opposite right" A: The output of "turn opposite
   right" concatenates: the output of "turn right", the output of "
   turn right". "turn right" outputs "TURN RIGHT". So repeating the
    output of "turn right" twice leads to "TURN RIGHT" * 2. So the
   output of "turn opposite right" is "TURN RIGHT" * 2.

Q: "turn around right"
```

```
A: The output of "turn around right" concatenates: the output of "
    turn right", the output of "turn right", the output of "turn
    right", the output of "turn right". "turn right" outputs "TURN
    RIGHT". So repeating the output of "turn right" four times leads
     to "TURN RIGHT" * 4. So the output of "turn around right" is "
    TURN RIGHT" * 4.

Q: "walk opposite left"
A: The output of "walk opposite left" concatenates: the output of "
    turn opposite left", the output of "walk". "turn opposite left"
    outputs "TURN LEFT" * 2. "walk" outputs "WALK". So concatenating
     the output of "turn opposite left" and the output of "walk"
    leads to "TURN LEFT" * 2 + "WALK". So the output of "walk
    opposite left" is "TURN LEFT" * 2 + "WALK".

Q: "walk around left"
A: The output of "walk around left" concatenates: the output of "
    walk left", the output of "walk left", the output of "walk left
    ", the output of "walk left". "walk left" outputs "TURN LEFT" +
    "WALK". So repeating the output of "walk around left" four times
     leads to ("TURN LEFT" + "WALK") * 4. So the output of "walk
    around left" is ("TURN LEFT" + "WALK") * 4.

Please pay attention to the use of parentheses.

Now, the first several sub-instructions are already solved, these
    sub-instructions are listed below following the sequence:{
    previous_steps}.

Their answers are listed below, also following the sequence:{
    previous_answs}.

Now you need to solve the sub-instruction: {Step_dict[str(cnt)]}.

Focus exclusively on solving the sub-instruction.
Your answer should be concise and directly address the core
    reasoning process.
Avoid any unnecessary comments, greetings, or expressions of
    enthusiasm. Only provide the essential reasoning process and
    answer.

Please provide the answer to the sub-instruction.
```

Below is the prompt for synthesizing to obtain the final answer on benchmark SCAN:

```
There is a natural language instruction representing a sequence of
    actions. I need you to translate this sentence from natural
    language into a standardized meta-action sequence."
Here is the instruction:{problem}

For your reference, 13 examples for translation together with the
    corresponding explanations are as follows:

Q: "turn left"
A: "turn left" outputs "TURN LEFT".

Q: "turn right"
A: "turn right" outputs "TURN RIGHT".
```

```
Q: "jump left"
A: The output of  jump   l e f t  concatenates: the output of  turn
     l e f t , the output of   jump   .  turn   l e f t  outputs
     TURN   L E F T .   jump   outputs   JUMP  . So concatenating
    the output of  turn   l e f t  and the output of   jump
   leads to  TURN   L E F T  +   JUMP   . So the output of  jump
    l e f t  is  TURN   L E F T  +   JUMP   .

Q: "run right"
A: The output of "run right" concatenates: the output of "turn right
   ", the output of "run". "turn right" outputs "TURN RIGHT". "run"
    outputs "RUN". So concatenating the output of "turn right" and
   the output of "run" leads to "TURN RIGHT" + "RUN". So the output
    of "run right" is "TURN RIGHT" + "RUN".

Q: "look twice"
A: The output of "look twice" concatenates: the output of "look",
   the output of "look". "look" outputs "LOOK". So repeating the
   output of "look" two times leads to "LOOK" * 2. So the output of
    "look twice" is "LOOK" * 2.

Q: "run and look twice"
A: The output of "run and look twice" concate+nates: the output of "
   run", the output of "look twice". "run" outputs "RUN". "look
   twice" outputs "LOOK" * 2. So concatenating the output of "run"
   and the output of "look twice" leads to "RUN" + "LOOK" * 2. So
   the output of "run and look twice" is "RUN" + "LOOK" * 2.

Q: "jump right thrice"
A: The output of "jump right thrice" concatenates: the output of "
   jump right", the output of "jump right", the output of "jump
   right". "jump right" outputs "TURN RIGHT" + "JUMP". So repeating
    the output of "jump right" three times leads to ("TURN RIGHT" +
    "JUMP") * 3. So the output of "jump right thrice" is ("TURN
   RIGHT" + "JUMP") * 3.

Q: "walk after run"
A: The output of "walk after run" concatenates: the output of "run",
    the output of "walk". "run" outputs "RUN". "walk" outputs "WALK
   ". So concatenating the output of "run" and the output of "walk"
    leads to "RUN" + "WALK". So the output of "walk after run" is "
   RUN" + "WALK".

Q: "turn opposite left"
A: The output of "turn opposite left" concatenates: the output of "
   turn left", the output of "turn left". "turn left" outputs "TURN
    LEFT". So repeating the output of "turn left" twice leads to "
   TURN LEFT" * 2. So the output of "turn opposite left" is "TURN
   LEFT" * 2.

Q: "turn around left"
A: The output of "turn around left" concatenates: the output of "
   turn left", the output of "turn left", the output of "turn left
   ", the output of "turn left". "turn left" outputs "TURN LEFT".
    So repeating the output of "turn left" four times leads to "TURN
    LEFT" * 4. So the output of "turn around left" is "TURN LEFT" *
    4. Q: "turn opposite right" A: The output of "turn opposite
   right" concatenates: the output of "turn right", the output of "
   turn right". "turn right" outputs "TURN RIGHT". So repeating the
    output of "turn right" twice leads to "TURN RIGHT" * 2. So the
   output of "turn opposite right" is "TURN RIGHT" * 2.

Q: "turn around right"
```

```
A: The output of "turn around right" concatenates: the output of "
   turn right", the output of "turn right", the output of "turn
   right", the output of "turn right". "turn right" outputs "TURN
   RIGHT". So repeating the output of "turn right" four times leads
    to "TURN RIGHT" * 4. So the output of "turn around right" is "
   TURN RIGHT" * 4.

Q: "walk opposite left"
A: The output of "walk opposite left" concatenates: the output of "
   turn opposite left", the output of "walk". "turn opposite left"
   outputs "TURN LEFT" * 2. "walk" outputs "WALK". So concatenating
    the output of "turn opposite left" and the output of "walk"
   leads to "TURN LEFT" * 2 + "WALK". So the output of "walk
   opposite left" is "TURN LEFT" * 2 + "WALK".

Q: "walk around left"
A: The output of "walk around left" concatenates: the output of "
   walk left", the output of "walk left", the output of "walk left
   ", the output of "walk left". "walk left" outputs "TURN LEFT" +
   "WALK". So repeating the output of "walk around left" four times
    leads to ("TURN LEFT" + "WALK") * 4. So the output of "walk
   around left" is ("TURN LEFT" + "WALK") * 4.

Please pay attention to the use of parentheses.

Following the order of the sub-instructions and solving every sub-
   instruction in sequence lead to the final answer.
All the sub-instructions are listed in the order: {previous_tasks}
The answers to all the sub-instructions are listed in the same order
   : {previous_answs}

We can synthesize the final answer based on all the answers to the
   sub-instructions.
You must synthesize the final answer strictly based on the provided
   answers to the sub-instructions, without performing any error
   correction or independent recalculations.
Even if a sub-instruction answer contains a reasoning mistake or
   calculation error, you must still use it as given.
Do not infer the correct answer based on correct reasoning steps if
   the computed result is incorrect.
Your final synthesis should reflect the exact values and conclusions
    stated in the sub-instruction answers, even if they are
   incorrect.

Please give the final action sequence without any additional
   explanation or clarification.
```

Below is the prompt for translating a pseudo action sequence expression into a sequence of actions on benchmark SCAN:

```
Now I have a pseudo action sequence expression with parentheses and
   multiplication. I need you to help me convert this into a
   sequence of actions without an operator sign.
6 examples are as follows:

Q: "JUMP" * 3
Rewrite: "JUMP" * 3
A: 1 JUMP 2 JUMP 3 JUMP
```

```
Q: "RUN" * 4 * 2
Rewrite: "RUN" * 8
A: 1 RUN 2 RUN 3 RUN 4 RUN 5 RUN 6 RUN 7 RUN 8 RUN

Q: "TURN RIGHT" + "WALK"
Rewrite: "TURN RIGHT" + "WALK"
A: TURN RIGHT WALK

Q: ("TURN LEFT" + "LOOK") * 2 + "TURN LEFT" + "LOOK"
Rewrite: ("TURN LEFT" + "LOOK") * 2 + "TURN LEFT" + "LOOK"
A: 1 (TURN LEFT LOOK) 2 (TURN LEFT LOOK) TURN LEFT LOOK

Q: ("TURN RIGHT" * 2 + "JUMP") * 4
Rewrite: ("TURN RIGHT" * 2 + "JUMP") * 4
A: 1 (1 TURN RIGHT 2 TURN RIGHT JUMP) 2 (1 TURN RIGHT 2 TURN RIGHT
    JUMP) 3 (1 TURN RIGHT 2 TURN RIGHT JUMP) 4 (1 TURN RIGHT 2 TURN
    RIGHT JUMP)

Q: "TURN LEFT" * 2 + ("TURN RIGHT" + "WALK") * 4 * 2
Rewrite: "TURN LEFT" * 2 + ("TURN RIGHT" + "WALK") * 8
A: 1 TURN LEFT 2 TURN LEFT 1 (TURN RIGHT WALK) 2 (TURN RIGHT WALK) 3
     (TURN RIGHT WALK) 4 (TURN RIGHT WALK) 5 (TURN RIGHT WALK) 6 (
    TURN RIGHT WALK) 7 (TURN RIGHT WALK) 8 (TURN RIGHT WALK)

The pseudo action sequence to be converted is as follows: {sentence}
Please change it to the action sequences.
Please JUST answer the result.
```

Below is the prompt for using task decomposer to decompose the original command into a sequence of sub-commands on benchmark SCAN:

```
I will give you a piece of natural language command. I need you to
    decompose it to smaller commands.

3 examples are as follows:

Command: "look right after look twice"
Answer:
The given command can finally be solved by: "look right", "look
    twice".

Command: "jump opposite right thrice and walk"
Answer:
The given command can finally be solved by: "jump opposite right", "
    jump opposite right thrice", "walk".

Command: "run left twice and run right"
Answer:
The given command can finally be solved by: "run left", "run left
    twice", "run right".

Now the command is {original_question}, please decompose it into
    smaller commands like the examples.
Answer Format:
The given command can finally be solved by: "subcommand_0", "
    subcommand_1",...
```

Below is the prompt for using subtask allocator to allocate candidate model to a certain sub-command on benchmark SCAN:

```
Now we have an original command.
To conduct this command, we decide to break this command into
    subcommands.
Now we have to conduct a subcommand, and there are 9 models that can
     be chosen to conduct this subcommand.
These 9 models are:
qwen2.5-0.5b, qwen2.5-1.5b, qwen2.5-3b, qwen2.5-7b, qwen2.5-14b,
    qwen2.5-32b, qwen2.5-72b, deepseek-V3, gpt-4o.
We list these models in ascending order acoording to their
    capability and the difficulty levels of the subcommands they are
     suitable for.
For example,
qwen2.5-0.5b has the lowest capability thus is suitable for the
    easiest subcommand
gpt-4o has the highest capability thus is suitable for the hardest
    subcommand.
Task: choose the most appropriate model from the list above to
    conduct the given subcommand.
Output only the chosen model's name.

the original command: {original_problem}
the subcommand: {subtask}
```

### E.3   SOLVING MATH PROBLEMS: MATH

Below is the prompt for decomposition data collection on benchmark MATH:

```
I will now give you a math problem. The type of problem is {type}.
    Please break this math problem down into several easy-to-solve
    steps.

These steps are organized in a chain-like manner, in which the steps
     are supposed to be solved following a certain order.
Meanwhile when writing each broken-down step, the order of the steps
     should be the order of how to solve these broken-down question
    steps.

1 examples are as follows:
Question: Four years ago, Kody was only half as old as Mohamed. If
    Mohamed is currently twice 30 years old, how old is Kody
    currently?
Answer: To solve the question "How old is Kody currently?", we need
    to know: "How old is Mohamed cuurently?", "How old was Mohamed
    four years ago?", "How old was Kody four years ago?".

Now the command is {question}, please decompose it into easy-to-
    solve steps like the examples.
Answer Format: (Please write each broken-down question step on a
    separate line, starting with a number.)
To solve the question "xxx", we need to know:
"1. question step 1",
"2. question step 2",
"3. question step 3".
...
```

Below is the prompt for solving sub-problems sequentially on benchmark MATH:

```
You are provided with a math problem. Your task is to solve it and
    give it an answer.
Here is the problem:\n{problem}
The question belongs to the type pf {question_type}.

I have broken this problem down into many easier subproblems.
Following the order of the subproblems and solving every subproblem
    in sequence lead to the final answer.

Now, the first several subproblems are already solved, these
    subproblems are listed below following their order:{
    previous_tasks}.
Their answers are listed below, also following their order:{
    previous_answs}.
Now you need to solve the subproblem: {Step_dict[str(cnt)]}.

Focus exclusively on solving the subproblem.
Your answer should be concise and directly address the core
    reasoning process.
Avoid any unnecessary comments, greetings, or expressions of
    enthusiasm. Only provide the essential reasoning process and
    answer.

Please provide the answer to the subproblem.
```

Below is the prompt for synthesizing to obtain the final answer on benchmark MATH:

```
We are provided with a math problem. Our task is to solve it and
    give it an answer.
Here is the problem:\{problem}
The question belongs to the type pf {question_type}.

I have broken this problem down into many easier subproblems.
Following the order of the subproblems and solving every subproblem
    in sequence lead to the final answer.
All the subproblems are listed in the order: {previous_tasks}
The answers to all the subproblems are listed in the same order: {
    previous_answs}

We can synthesize the final answer based on all the answers to the
    subproblems.
You must synthesize the final answer strictly based on the provided
    answers to the subproblems, without performing any error
    correction or independent recalculations.
Even if a subproblem answer contains a reasoning mistake or
    calculation error, you must still use it as given.
Do not infer the correct answer based on correct reasoning steps if
    the computed result is incorrect.
Your final synthesis should reflect the exact values and conclusions
     stated in the subproblem answers, even if they are incorrect.

Please give the final answer without any additional explanation or
    clarification.
```

Below is the prompt for judging if the final answer is correct on benchmark MATH:

```
Here is a math problem with a standard answer and a student's answer
    . Please help me determine if the student's answer is correct.
Problem: {problem}

question type: {question_type}

Standard answer: {solution}

Answer: {final_answ}

If the student's answer is correct, just output True; otherwise,
    just output False.
No explanation is required.
```

Below is the prompt for using task decomposer to decompose the original problem into a sequence of sub-problems on benchmark MATH:

```
I will now give you a math problem. Please break this math problem
    down into several easy-to-solve steps.

These sub-problems are supposed to be solved in a chain-like manner
    following a certain order.
When writing each broken-down sub-problem, the order of the sub-
    problems should be the order of solving these broken-down sub-
    problems.

1 examples is as follows:
Question: Four years ago, Kody was only half as old as Mohamed. If
    Mohamed is currently twice 30 years old, how old is Kody
    currently?
Answer:
To solve the given question, we need to know:
1. How old is Mohamed cuurently?
2. How old was Mohamed four years ago?
3. How old was Kody four years ago?

Now the command is {original_question}, please decompose it into
    easy-to-solve steps like the examples.
Answer Format: (Please write each broken-down question step on a
    separate line, starting with a number.)
To solve the given question, we need to know:
1. question step 1
2. question step 2
3. question step 3
...
```

Below is the prompt for using subtask allocator to allocate candidate model to a certain sub-problem on benchmark MATH:

```
Now we have an original problem.
To solve this problem, we decide to break this problem into
    subproblems.
Now we have to solve a subproblem, and there are 9 models that can
    be chosen to solve this subproblem.
These 9 models are:
```

```
qwen2.5-0.5b, qwen2.5-1.5b, qwen2.5-3b, qwen2.5-7b, qwen2.5-14b,
    qwen2.5-32b, qwen2.5-72b, deepseek-V3, gpt-4o.\n\n
We list these models in ascending order acoording to their
    capability and the difficulty levels of the subproblems they are
     suitable for.
For example,
qwen2.5-0.5b has the lowest capability thus is suitable for the
    easiest subproblem
gpt-4o has the highest capability thus is suitable for the hardest
    subproblem.
Task: choose the most appropriate model from the list above to solve
     the given subproblem.
Output only the chosen model's name.

the original problem: {original_problem}
the subproblem: {subtask}
```

### E.4 COMMONSENSE REASONING: CSQA

Below is the prompt for decomposition data collection on benchmark CSQA:

```
I have a single-choice question involving common sense reasoning
    that I want to solve. I hope you can break down the problem-
    solving process into several sub-problems. You can consider
    analyzing the question itself as well as the options.
The number of sub-problems doesn't need to be too many; each sub-
    problem should have a clear meaning and purpose.

These sub-problems are organized in a chain-like manner, in which
    the sub-problems are supposed to be solved following a certain
    order.
Meanwhile when writing each broken-down sub-problem, the order of
    the sub-problems should be the order of how to solve these
    broken-down sub-problems.

8 examples are as follows:
Question 1:
You can read a magazine where while waiting for your transportation
    on rails to arrive?
Choices 1:
A. Train station, B. Bookstore, C. Newsstand, D. Waiting room, E.
    Airport
Answer 1:
1. What does "waiting for your transportation on rails" indicate
    about your current location?
2. Which place in the options can accommodate you reading a magazine
    ?
3. Which places that satisfy question 2 are near your current
    location?

Question 2:
If I wanted to see a lizard in its natural habitat but I do not
    speak Spanish, where would I go?
Choices 2:
A. Utahc, B. South America, C. New Hampshire, D. Japan, E. New
    Mexico
Answer 2:
1. Which places are natural habitats for lizards?
2. Which places have Spanish as the primary language?
```

```
3. Combine the answers from sub-question 1 and sub-question 2, among
     the natural habitats for lizards, which places do not speak
     Spanish?

Question 3:
John was stuck in his house.  He couldn't get out the door.  He was
     very frightened when the smoke detectors went off, but luckily
     it was a false alarm.  Why might he be stuck?
Choices 3:
A. fire,  B. belong to, C. winter storm, D.face south, E.burn down
Answer 3:
1. What are possible reasons for being stuck in a house?
2. Which options are related to situations that might cause a person
     to be stuck?
3. Why might these specific conditions make it difficult to leave
     the house?

Question 4:
John was stuck in his house.  He couldn't get out the door.  He was
     very frightened when the smoke detectors went off, but luckily
     it was a false alarm.  Why might he be stuck?
Choices 4:
A. fire,  B. belong to, C. winter storm, D.face south, E.burn down
Answer 4:
1. What are possible reasons for being stuck in a house?
2. Which options are related to situations that might cause a person
     to be stuck?
3. Why might these specific conditions make it difficult to leave
     the house?

Question 5:
When looking for a non-perishable food in your house, you'll often
     go look in the?
Choices 5:
A. Stove, B. Table, C. Plate, D. Jar, E. Pantry
Answer 5:
1. What is non-perishable food?
2. Where are non-perishable foods commonly stored in a household?
3. Which of the options (stove, table, plate, jar, pantry) is the
     most logical place for storing non-perishable food?

Question 6:
What must elementary school students do when they are going into
     class?
Choices 6:
A. Think for himself, B. Answer question, C. Wait in line, D. Speak
     a foreign language, E. Cross road
Answer 6:
1. What do elementary school students typically do before entering a
     classroom?
2. Which actions among the options are related to classroom entry
     procedures?
3. Why might students perform this action before entering the
     classroom?

Question 7:
After eating dinner, having plenty to eat, and exercising, what is
     likely to happen?
Choices 7:
A. Become tired, B. Indigestion, C. Flatulence, D. Become
     intoxicated, E. Become full
Answer 7:
1. What happens to the body after eating a large meal?
```

```
2. What are common effects of exercising after eating?
3. Which of the options (become tired, indigestion, flatulence,
   become intoxicated, become full) best matches the expected
   outcome of eating a large meal followed by exercise?

Question 8:
He didn't like the walk up, but living on the top floor meant there
    was nobody above him in the what?
Choices 8:
A. Apartment building, B. Tall building, C. Go down, D. Garden, E.
    Office building
Answer 8:
1. What does "walk up" suggest about the type of building?
2. What kind of building would have a "top floor" and residents
    living above or below each other?
3. Which option (apartment building, tall building, go down, garden,
    office building) best describes a place where living on the top
    floor would mean no one lives above?

Now the question is {question}, the options are: {options}, please
    decompose it into sub-problems.
Answer Format: (Please write each broken-down sub-problem on a
    separate line, starting with a number.)
To solve the question "xxx", we need to clarify / solve:
"1. sub-problem 1",
"2. sub-problem 2",
"3. sub-problem 3".
...
```

Below is the prompt for solving sub-problems sequentially on benchmark CSQA:

```
There is a single-choice question involving common sense reasoning.
    I need you to solve it and give the right answer.
Here is the question:{problem}
Here are the options:{options}

I have broken this common sense reasoning question down into several
     smaller subproblems.
Following the order of the subproblems and solving every subproblem
    in sequence lead to the final answer.

Now, the first several subproblems are already solved, these
    subproblems are listed below following their order:{
    previous_tasks}.
Their answers are listed below, also following their order:{
    previous_answs}.
Now you need to solve the subproblem: {Step_dict[str(cnt)]}.

Focus exclusively on solving the subproblem.
Your answer should be concise and directly address the core
    reasoning process.
Avoid any unnecessary comments, greetings, or expressions of
    enthusiasm. Only provide the essential reasoning process and
    answer.

Please provide the answer to the subproblem.
```

Below is the prompt for synthesizing to obtain the final answer on benchmark CSQA:

```
There is a single-choice question involving common sense reasoning.
    I need you to solve it and give the right answer.
Here is the question:{problem}
Here are the options:{options}

I have broken this common sense reasoning question down into several
    smaller subproblems.
Following the order of the subproblems and solving every subproblem
    in sequence lead to the final answer.
All the subproblems are listed in the order: {previous_tasks}
The answers to all the subproblems are listed in the same order: {
    previous_answs}

We can synthesize the final answer based on all the answers to the
    subproblems, and finally choose the letter of the correct option
    .
You must synthesize the final answer strictly based on the provided
    answers to the subproblems, without performing any error
    correction or independent recalculations.
Even if a subproblem answer contains a reasoning mistake or
    calculation error, you must still use it as given.
Do not infer the correct answer based on correct reasoning steps if
    the computed result is incorrect.
Your final synthesis should reflect the exact values and conclusions
    stated in the subproblem answers, even if they are incorrect.

Please provide only the letter of the option, without any additional
    explanation or description.
```

Below is the prompt for using task decomposer to decompose the original problem into a sequence of sub-problems on benchmark CSQA:

```
I have a single-choice question involving common sense reasoning
    that I want to solve. I hope you can break down the problem-
    solving process into several sub-problems. You can consider
    analyzing the question itself as well as the options.
The number of sub-problems doesn't need to be too many; each sub-
    problem should have a clear meaning and purpose.

These sub-problems are supposed to be solved in a chain-like manner
    following a certain order.
When writing each broken-down sub-problem, the order of the sub-
    problems should be the order of solving these broken-down sub-
    problems.

1 example is as follows:
Question 1:
You can read a magazine where while waiting for your transportation
    on rails to arrive?
Choices 1:
A. Train station, B. Bookstore, C. Newsstand, D. Waiting room, E.
    Airport
Answer 1:
To solve the given question, we need to clarify / solve:
1. What does "waiting for your transportation on rails" indicate
    about your current location?
```

```
2. Which place in the options can accommodate you reading a magazine
    ?
3. Which places that satisfy question 2 are near your current
    location?

Now the question is {original_question}, the options are: {options},
     please decompose it into sub-problems.
Answer Format: (Please write each broken-down sub-problem on a
    separate line, starting with a number.)
To solve the given question, we need to clarify / solve:
1. sub-problem 1,
2. sub-problem 2,
3. sub-problem 3.
...
```

Below is the prompt for using subtask allocator to allocate candidate model to a certain sub-problem on benchmark CSQA:

```
Now we have an original problem.
To solve this problem, we decide to break this problem into
    subproblems.
Now we have to solve a subproblem, and there are 9 models that can
    be chosen to solve this subproblem.
These 9 models are:
qwen2.5-0.5b, qwen2.5-1.5b, qwen2.5-3b, qwen2.5-7b, qwen2.5-14b,
    qwen2.5-32b, qwen2.5-72b, deepseek-V3, gpt-4o.\n\n
We list these models in ascending order acoording to their
    capability and the difficulty levels of the subproblems they are
     suitable for.
For example,
qwen2.5-0.5b has the lowest capability thus is suitable for the
    easiest subproblem
gpt-4o has the highest capability thus is suitable for the hardest
    subproblem.
Task: choose the most appropriate model from the list above to solve
     the given subproblem.
Output only the chosen model's name.

the original problem: {original_problem}
the subproblem: {subtask}
```

You may include other additional sections here.

