# OpenReview forum: "Route-and-Reason: Scaling Large Language Model Reasoning with Reinforced Model Router"
_ICLR.cc/2026/Conference — ICLR 2026 Conference Withdrawn Submission_

### Official Review · Reviewer_Uy3x · 2025-10-26

**Soundness:** 2
**Presentation:** 2
**Contribution:** 2
**Rating:** 2
**Confidence:** 4

**Summary:**

The paper proposes R2‑Reasoner, a framework that decomposes a complex query into subtasks and then routes each subtask to an appropriately-sized model (from lightweight small-language models (SLMs) to full large language models (LLMs)), via a “Model Router” composed of (i) a Task Decomposer and (ii) a Subtask Allocator. Training of the router is via supervised fine-tuning followed by an iterative reinforcement‐learning pipeline using a Group Relative Policy Optimization (GRPO) style algorithm. The goal is to save inference cost (API or compute) by using smaller models when possible, while retaining accuracy

**Strengths:**

- The framework brings together (a) subtask decomposition, (b) difficulty estimation/allocation, and (c) reinforcement fine‐tuning of the router. While decomposition and routing have been studied, combining them in a staged RL training pipeline is interesting. The “Model Router” concept is clearly described.
- The reported results (cost reduction ~86.85%) are impressive on the face of it. The authors show that their method achieves much lower API cost while maintaining accuracy in the benchmarks they use.

**Weaknesses:**

- What exactly is *Meval* — a baseline evaluation model or a practical deployment reference? The paper states that Meval is used to estimate *practicality*, but does not specify its configuration. Given that different reasoning models generate reasoning traces of varying lengths, the selection of *Meval* could directly influence the reported API cost of R2-Reasoner. Clarification is also needed on whether *Meval* is an  edge or  cloud-based model, since this has implications for both latency and cost.
- The weights ( w_c, w_p, w_d ) in Equation (1) are introduced but not explained — how are they chosen, tuned, or justified? Similarly, the term ( \text{Coe}_{pair} ) is undefined; how is it computed, and what is its role in the data generation process? These details are crucial for reproducing the Task Decomposer’s training data and for understanding the overall scoring formulation.
- Referring to the statement at line 212 and Equation (7), the procedure for selecting difficulty thresholds ( \tau ) is unclear. What values of ( \alpha ) are used for the α-quantile token probability, and how exactly are the ( p(t_i) ) values obtained? Are they the predicted probabilities of the final token in each subtask, or averaged across all tokens? These implementation details materially affect how subtask difficulty is estimated.
- I also doubt the claimed latency improvements. In practice, such routing approaches likely **increase latency** due to model-switching overhead and KV-cache inefficiency. Each time the router switches from one model to another (e.g., from a small to a large LLM), the system must re-encode the full prompt and regenerate a new key–value (KV) cache, since caches are model-specific and incompatible across architectures. This eliminates the usual ( O(n) ) decoding benefit of KV caching and reintroduces ( O(n^2) ) attention computation for every routed subtask. Consequently, wall-clock latency could increase substantially — potentially 2–3× for multi-model chains — even if token-level cost decreases by 80 % or more.
- The results in Table 3 clearly indicate that when the model pool is modified (e.g., replacing or updating one of the base models), both accuracy and cost efficiency degrade noticeably even under identical experimental conditions. This suggests that the router and decomposer are tightly coupled to the exact composition of the model pool used during training. Consequently, the framework appears fragile to model updates: whenever a model in the pool is replaced, fine-tuned, or version-upgraded, the router would likely require retraining or recalibration, incurring substantial additional cost. This undermines the claim of practical efficiency — it is not a one-time investment but an ongoing maintenance expense. The paper should discuss whether incremental adaptation or lightweight retraining strategies are feasible to mitigate this dependency.

**Questions:**

- Line 196 introduces ( d^* ) without any definition. Although Appendix B.1 later provides it, appendices should not be required to follow the main argument — this breaks the continuity of the reader. Likewise, ( M^* ) appears at line 207 with no immediate explanation. These omissions reduce clarity in the core methodology presentation.
- In Equation (6), are ( C_{api} ) and ( C_{time} ) on the same numerical scale? If not, one term will dominate the other during optimization. The paper should clarify how these quantities are normalized or balanced to ensure a meaningful joint objective.
-  Line 215 introduces the segmentation of models into SLM, MLM, and LLM categories, but the criteria for this division are not described. Are these based on parameter count, training data size, or empirical performance? Furthermore, linking back to *Meval*, it is unclear which group *Meval* belongs to.

---

### Official Review · Reviewer_sQpA · 2025-10-29

**Soundness:** 3
**Presentation:** 3
**Contribution:** 3
**Rating:** 6
**Confidence:** 3

**Summary:**

R2-Reasoner is a framework for efficient large language model (LLM) reasoning, motivated by the high computational costs of chain-of-thought (CoT) reasoning and the limitations of existing model routers that operate at coarse, task-level granularity. It introduces a Reinforced Model Router that decomposes complex queries into subtasks and strategically allocates them across heterogeneous models.

Key innovations include: (1) a two-stage training pipeline combining supervised fine-tuning (SFT) and reinforcement learning (RL); (2) fine-grained, subtask-level collaboration among models; and (3) an 84.46% reduction in API costs while maintaining competitive—sometimes improved—reasoning accuracy across six challenging benchmarks.

**Strengths:**

**Originality**

Proposes one of the first frameworks to enable subtask-level routing with an alternating RL training scheme that co-optimizes decomposition and allocation.

**Quality**

Rigorous evaluation across six benchmarks (P3, SCAN, MATH, CHAMP, CSQA, MuSiQue) shows strong cost–accuracy trade-offs. Ablation studies support the value of the SFT+RL pipeline, with RL improving proxy measures of subtask quality and reducing allocation errors.

**Clarity & Significance**

Well-structured and clearly articulated. The framework could facilitate cost-efficient LLM deployment in edge-cloud or real-time settings.

**Weaknesses:**

**Decomposition Evaluation**

The reported 27% improvement in “subtask correctness” relies on automatic proxy metrics (e.g., coherence), not human-annotated gold standards, limiting confidence in the true reasoning fidelity.

**Synthetic Training Data**

The decomposer is trained on synthetically generated CoT traces, which may not reflect the diversity or nuance of human reasoning, risking brittleness on out-of-distribution queries.

**Sparse Reward Signal**

The binary final-answer reward provides minimal credit assignment for intermediate decisions, potentially slowing RL convergence or trapping policies in local optima.

**Questions:**

1. How does the framework handle potential knowledge inconsistencies across heterogeneous models (e.g., differing factual beliefs), which could introduce contradictions in multi-step reasoning?

2. While subtask routing reduces per-step errors, how does R2-Reasoner mitigate error compounding when early subtask mistakes propagate through subsequent reasoning steps?

---

### Official Review · Reviewer_xbqF · 2025-11-01

**Soundness:** 2
**Presentation:** 2
**Contribution:** 1
**Rating:** 4
**Confidence:** 3

**Summary:**

The paper targets at lowering down the inference cost for the reasoning steps. Instead of always calling the strong but expensive solver model, the authors argue that routing to a small model to resolve the easy sub-task is a much smarter way. To achieve this, the authors propose to train a task decomposer and a sub-task router model together. They first construct two datasets to SFT the model. And then iteratively do the RL training for decomposer and router.

The authors show the strong results on six reasoning tasks.

**Strengths:**

1. The shown results are solid.
2. Most claims are supported by results.

**Weaknesses:**

1. The authors highlights the gain in reducing inference cost while keeping a comparable (or better) performance on six reasoning tasks. However, training cost is not mentioned. As the training produce leverages a strong model as reward to refine the weak model's reward, I assume this would become a reasonable high cost.
2. The paper does not show a training process, it would be more helpful to provide training process in pictures.
3. It would be more helpful to extend the ablation studies section to include the answer for below questions.

**Questions:**

1. Why iterative train the decomposer and allocator in RL?
2. What makes RL work? Given that the only signal comes from final verifiable result, what guides the intermedia process change?
3. The Procedural Review Mechanism requests a call of strong but costy model for each subtask, will this setting make the training too expensive?
4. What will happen if the we collect all correct/refined reward for each r_j and re-finetune the assigned model using this supervision signals? Will this method stronger than the proposed RL pipeline?
5. The rollout size is 4 which seems too small to make the RL training work with 1024 training steps. Can you provide corresponding training process?

---

### Official Review · Reviewer_gJ4a · 2025-11-01

**Soundness:** 2
**Presentation:** 2
**Contribution:** 2
**Rating:** 4
**Confidence:** 4

**Summary:**

This paper introduces R2-Reasoner, a novel framework that employs a reinforced model router to efficiently scale LLM reasoning by decomposing complex queries into subtasks and strategically allocating them across heterogeneous models. The system uses a two-stage training process combining supervised fine-tuning with reinforcement learning, achieving an 84.46% reduction in API costs while maintaining competitive accuracy across six reasoning benchmarks. The approach represents a shift from task-level to subtask-level model routing, enabling finer-grained collaboration between models of varying capabilities.

**Strengths:**

1. The framework demonstrates remarkable cost reduction (84.46%) while maintaining or even improving accuracy on several benchmarks, addressing a critical practical concern in deploying LLM reasoning systems at scale.
2. By operating at the subtask level rather than the task level, the approach enables more efficient utilization of model capabilities, matching computational resources to actual complexity requirements of individual reasoning steps.
3. The paper provides extensive experiments across six diverse benchmarks with detailed ablation studies, demonstrating robustness and includes practical considerations like generalization to unseen models and controllable accuracy-cost trade-offs.

**Weaknesses:**

1. Figure 1 fails to effectively communicate the method's core logic, particularly lacking clear visual distinction between the Downward Allocation and Upward Allocation processes, making it difficult to understand the iterative refinement strategy at a glance.
2. The use of maximum token probability as a proxy for task difficulty raises concerns about validity. The paper lacks justification for the specific threshold values (τ_easy, τ_medium, τ_hard) and doesn't clearly explain how the baseline model's difficulty assessment maps to the model capability groups.
3. The framework's heavy reliance on initial task decomposition quality may limit its effectiveness on more challenging datasets like AIME or GPQA, where problems require complex self-verification and reflection that cannot be reliably decomposed upfront.
4. The paper shows R2-Reasoner outperforming GPT-4o on certain benchmarks despite using it as an upper-bound model with PRM verification, but lacks detailed analysis of which specific problems exhibit this behavior and why.

**Questions:**

1. Does the system allocate models based on fixed ordinal positions? When models are added or removed from the pool, does the entire framework require retraining or can it adapt dynamically?
2. How exactly is the relative ordering of model capabilities established? Is this based on benchmark performance, parameter count, or some other metric?
3. What proportion of subtasks trigger the Procedural Review Mechanism? This metric would help understand the actual cost-performance trade-off in practice.
4. Table 8 shows decreasing token consumption as model capability increases (averaging only 90 tokens for MATH), which seems counterintuitively low. Is this due to the selected subset being relatively simple? Could you evaluate on more challenging subsets like MATH500?
5. How would R2-Reasoner adapt to problems requiring iterative refinement or self-correction during the reasoning process, rather than linear subtask execution?

---

### Note · Authors · 2025-11-26

I have read and agree with the venue's withdrawal policy on behalf of myself and my co-authors.